# NONPARAMETRIC TEACHING OF ATTENTION LEARNERS

**Chen Zhang**[1]* **Jianghui Wang**[2]* **Bingyang Cheng**[1] **Zhongtao Chen**[1] **Wendong Xu**[1]
**Cong Wang**[3] **Marco Canini**[2] **Francesco Orabona**[2] **Yik-Chung Wu**[1] **Ngai Wong**[1]

[1] The University of Hong Kong
[2] King Abdullah University of Science and Technology
[3] Independent Researcher
czhang6@connect.hku.hk   jianghui.wang@kaust.edu.sa

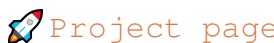 Project page

## ABSTRACT

Attention learners, neural networks built on the attention mechanism, *e.g.*, transformers, excel at learning the implicit relationships that relate sequences to their corresponding properties, *e.g.*, mapping a given sequence of tokens to the probability of the next token. However, the learning process tends to be costly. To address this, we present a novel paradigm named **Atte**ntion **N**eural **T**eaching (AtteNT) that reinterprets the learning process through a nonparametric teaching perspective. Specifically, the latter provides a theoretical framework for teaching mappings that are implicitly defined (*i.e.*, nonparametric) via example selection. Such an implicit mapping is embodied through a dense set of sequence-property pairs, with the AtteNT teacher selecting a subset to accelerate convergence in attention learner training. By analytically investigating the role of attention on parameter-based gradient descent during training, and recasting the evolution of attention learners, shaped by parameter updates, through functional gradient descent in nonparametric teaching, we show *for the first time* that teaching attention learners is consistent with teaching importance-adaptive nonparametric learners. These new findings readily commit AtteNT to enhancing learning efficiency of attention learners. Specifically, we observe training time reductions of 13.01% for LLMs and 20.58% for ViTs, spanning both fine-tuning and training-from-scratch regimes. Crucially, these gains are achieved without compromising accuracy; in fact, performance is consistently preserved and often enhanced across a diverse set of downstream tasks.

## 1 INTRODUCTION

The attention mechanism, inspired by human attention concepts (Ahmad, 1991; Soydaner, 2022), is designed to assess the relative importance of each element in a sequence (Bahdanau et al., 2015; Vaswani et al., 2017). By leveraging attention, neural networks can effectively learn the implicit relationships that map sequences to their corresponding properties, *e.g.*, mapping a sequence of tokens to the probability of the next token. These Attention Neural Networks (ANNs), *e.g.*, transformers (Vaswani et al., 2017; Kong et al., 2019), have achieved significant success in a wide range of downstream tasks across various fields, including natural language processing (Vaswani et al., 2017; Devlin et al., 2019; Consens et al., 2025), computer vision (Dosovitskiy et al., 2020; Azad et al., 2024; Chen et al., 2024), and multimodal systems (Nagrani et al., 2021; Yang et al., 2024).

However, the process of learning the implicit mappings (*i.e.*, training) can be quite costly for ANNs, especially when handling large-scale tasks (Liu et al., 2018b; Beltagy et al., 2019; Gu et al., 2021; Yang et al., 2023). For instance, pretraining language models often requires training on corpora with millions of sentences (Common Crawl, 2007; Li et al., 2024). In the case of video understanding, the

---

*Equal contribution

scale can become overwhelmingly large (Bain et al., 2021; Sharma et al., 2018; Shu et al., 2025). As a result, reducing training costs and enhancing learning efficiency has become an urgent priority.

Recent research on nonparametric teaching (Zhang et al., 2023b;a; 2024a; 2025) presents a promising solution to the issue outlined above. Specifically, nonparametric teaching provides a theoretical framework for selecting examples efficiently when the target mapping (*i.e.*, either a function or a model) being taught is nonparametric, *i.e.*, implicitly defined. It builds on the concept of machine teaching (Zhu, 2015; Zhu et al., 2018), which involves designing a training set (dubbed the teaching set) to help the learner quickly converge to the target functions, while relaxing the assumption that the target functions are parametric (Liu et al., 2017; 2018a), thus enabling the teaching of nonparametric (non-closed-form) functions with a focus on function space. Unfortunately, these studies are limited to multilayer perceptron-based learners and do not account for the attention mechanism, making their direct application difficult when the learners are ANNs. Additionally, ANNs are typically updated through gradient descent in parameter space, which contrasts with the functional gradient descent used in nonparametric teaching within function space (Zhang et al., 2023b;a; 2024a; 2025). Hence, it is not immediate to apply nonparametric teaching theory to attention learners.

To this end, we systematically investigate the role of attention on ANN gradient-based training in both parameter and function spaces. Specifically, we analytically examine how attention adaptively assigns different importance to each element in an input sequence during parameter-based gradient descent in parameter space, and explicitly show that the parameter gradient retains the same form as the input sequence size scales. This importance-adaptive update in parameter space drives the evolution of the ANN, which can be expressed using the dynamic Attention Neural Tangent Kernel (ANTK) (Yang, 2019; Hron et al., 2020), and then cast into function space. We prove that this dynamic ANTK converges to the importance-adaptive canonical kernel used in functional gradient descent, suggesting that the evolution of ANN under parameter gradient descent is consistent with that under functional gradient descent. Therefore, it is natural to interpret the learning process of attention learners through the theoretical framework of nonparametric teaching: the target mapping is represented by a dense set of sequence-property pairs, where each sequence is associated with its target output, and the teacher selects a subset of these pairs to provide to the ANN, ensuring rapid convergence of this attention learner. Consequently, to improve the learning efficiency of ANNs, we propose a novel paradigm called AtteNT, where the teacher applies a counterpart of the greedy teaching algorithm from nonparametric teaching to train attention learner, specifically by selecting the sequence with the greatest discrepancy between their true property values and the ANN outputs. Lastly, we carry out comprehensive experiments to demonstrate the effectiveness of AtteNT across various scenarios, including both natural language processing and computer vision tasks. Our key contributions are as follows:

- We propose AtteNT, a novel paradigm that interprets attention learner training through the theoretical lens of nonparametric teaching. This facilitates the use of greedy algorithms from nonparametric teaching to effectively improve the learning efficiency of attention learners.

- We analytically investigate the role of attention in parameter-based gradient descent within parameter space, revealing the consistency between the evolution of ANN driven by parameter updates and that under functional gradient descent in nonparametric teaching. We further show that the dynamic ANTK, emerging from gradient descent on the parameters, converges to the importance-adaptive canonical kernel of functional gradient descent. These findings bridge nonparametric teaching theory with attention learner training, thereby broadening the application of nonparametric teaching to contexts involving attention mechanisms.

- We demonstrate the effectiveness of AtteNT through extensive experiments across both natural language processing (NLP) and computer vision (CV) tasks. Our approach reduces Large Language Model (LLM) fine-tuning time by 13.01% and accelerates Vision Transformer (ViT) training from scratch by 20.58%, thereby providing strong empirical support for our theoretical claims.

## 2 RELATED WORKS

**Attention learners**. The effectiveness of the attention mechanism in learning implicit mappings from sequences to relevant properties has spurred a surge in research on attention learners (Bahdanau et al., 2015; Vaswani et al., 2017; Kong et al., 2019). This growing interest is particularly evident in the

increasing efforts to apply attention learners across a wide variety of downstream tasks, including natural language processing (Galassi et al., 2020; Jin et al., 2024), computer vision (Dosovitskiy et al., 2020; Hassanin et al., 2024; Zhang et al., 2024b), medicine (Thirunavukarasu et al., 2023; Demszky et al., 2023), and graph-related fields (Veličković et al., 2018; Wu et al., 2024). Various efforts have been made in designing learners for improved mapping learning in vision tasks (Lin et al., 2022; Dosovitskiy et al., 2020; Arnab et al., 2021), as well as for more efficient inference (Kitaev et al., 2020; Katharopoulos et al., 2020; Lu et al., 2021). There have also been ongoing pursuits to enhance learning efficiency, such as sparse training (Frankle & Carbin, 2018; You et al., 2020; Chen et al., 2021c;b; Li et al., 2023), improved initialization (Huang et al., 2020; d'Ascoli et al., 2021), and data curation (Tang et al., 2023; Zhong et al., 2023; Lin et al., 2024; Li et al., 2024). Differently, we frame attention learner training from a fresh perspective of nonparametric teaching (Zhang et al., 2023b;a), and adopt a corresponding variant of the greedy algorithm to enhance the training efficiency of ANNs.

**Nonparametric teaching**. Machine teaching (Zhu, 2015; Zhu et al., 2018) focuses on designing a teaching set that allows the learner to quickly converge to a target model function. It can be seen as the reverse of machine learning: while machine learning aims to learn a mapping from a given training set, machine teaching seeks to construct the set based on a desired mapping. Its effectiveness has been demonstrated across various domains, including crowdsourcing (Singla et al., 2014; Zhou et al., 2018), robustness (Alfeld et al., 2017; Ma et al., 2019; Rakhsha et al., 2020), and computer vision (Wang et al., 2021a; Wang & Vasconcelos, 2021). Nonparametric teaching (Zhang et al., 2023b;a) extends iterative machine teaching (Liu et al., 2017; 2018a) by broadening the parameterized family of target mappings to encompass the more general nonparametric framework. This theoretical framework has proven effective in enhancing the efficiency of multilayer perceptrons for learning implicit functions from signal coordinates to corresponding values (Zhang et al., 2024a; 2026), as well as improving the training efficiency of graph convolutional networks for learning implicit mappings from graphs to their relevant properties (Zhang et al., 2025). Nevertheless, the absence of the attention mechanism in these studies limits their direct applicability to general tasks involving attention learners (Bahdanau et al., 2015; Vaswani et al., 2017). This work systematically investigates the role of attention and highlights the alignment between the evolution of ANN driven by parameter updates and that guided by functional gradient descent in nonparametric teaching. These insights, for the first time, broaden the scope of nonparametric teaching in attention learner training, positioning our AtteNT as a novel approach to improving ANN learning efficiency.

## 3 BACKGROUND

**Notation.**[1] Let $(\boldsymbol{x}_1, \ldots, \boldsymbol{x}_S)$ represent a sequence of length $S$, where each $\boldsymbol{x}_s \in \mathbb{R}^d$ denotes a $d$-dimensional feature vector associated with the $s$-th element, with $s \in \mathbb{N}_S$ ($\mathbb{N}_S \coloneqq \{1, \ldots, S\}$). Each $\boldsymbol{x}_s$ is a row vector, expressed as $[x_{s,j}]_d^\top = [x_{s,1}, \ldots, x_{s,d}]$. The entire collection of feature vectors forms an $S \times d$ feature matrix, denoted $\boldsymbol{S}_{S \times d} \in \mathcal{S} \subseteq \mathbb{R}^{S \times d}$ (or simply $\boldsymbol{S}$). The $s$-th row and the $i$-th column of this matrix, corresponding to the $s$-th element and the $i$-th feature, are denoted by $\boldsymbol{S}_{(s,:)}$ and $\boldsymbol{S}_{(:,i)}$, respectively. Alternatively, these can be written as $\boldsymbol{e}_s^\top \boldsymbol{S}$ and $\boldsymbol{S} \boldsymbol{e}_i$, where $\boldsymbol{e}_i$ is a standard basis vector with its $i$-th entry being 1 and all other entries equal to 0. The bold column vector $\mathbf{1}$ represents a vector in which all elements are 1. The property of the sequence is represented by $\boldsymbol{y} \in \mathcal{Y}$, where $\boldsymbol{y}$ is a scalar for sequence-level properties ($\mathcal{Y} \subseteq \mathbb{R}$) and a vector for element-level properties ($\mathcal{Y} \subseteq \mathbb{R}^n$). A set with $m$ items is denoted as $\{a_i\}_m$. If $\{a_i\}_m \subseteq \{a_i\}_n$, then $\{a_i\}_m$ represents a subset of $\{a_i\}_n$ containing $m$ items, where the indices are $i \in \mathbb{N}_n$. A diagonal matrix with diagonal entries $a_1, \ldots, a_m$ is denoted as $\mathrm{diag}(a_1, \ldots, a_m)$, and if all $m$ values are identical, the matrix is simplified as $\mathrm{diag}(a; m)$.

Let $K(\boldsymbol{S}, \boldsymbol{S}') : \mathcal{S} \times \mathcal{S} \mapsto \mathbb{R}$ denote a symmetric and positive definite sequence kernel (Cancedda et al., 2003; Király & Oberhauser, 2019). This kernel can also be expressed as $K(\boldsymbol{S}, \boldsymbol{S}') = K_{\boldsymbol{S}}(\boldsymbol{S}') = K_{\boldsymbol{S}'}(\boldsymbol{S})$, where for simplicity, $K_{\boldsymbol{S}}(\cdot)$ may be abbreviated as $K_{\boldsymbol{S}}$. The reproducing kernel Hilbert space (RKHS) $\mathcal{H}$ associated with $K(\boldsymbol{S}, \boldsymbol{S}')$ is defined as the closure of the linear span $\{f : f(\cdot) = \sum_{i=1}^r a_i K(\boldsymbol{S}_i, \cdot), a_i \in \mathbb{R}, r \in \mathbb{N}, \boldsymbol{S}_i \in \mathcal{S}\}$, with the inner product given by $\langle f, g \rangle_{\mathcal{H}} = \sum_{ij} a_i b_j K(\boldsymbol{S}_i, \boldsymbol{S}_j)$, where $g = \sum_j b_j K_{\boldsymbol{S}_j}$ (Liu & Wang, 2016; Zhang et al., 2023b). Rather than assuming the idealized case of a closed-form solution $f^*$, we focus on the more realistic scenario where the realization of $f^*$ is given (Zhang et al., 2023b;a; 2024a; 2025). Given the target

---

[1]The notation table can be found in Appendix A.1.

mapping $f^* : \mathcal{S} \mapsto \mathcal{Y}$, it uniquely maps each sequence $\boldsymbol{S}_\dagger$ to its corresponding output $\boldsymbol{y}_\dagger$, such that $\boldsymbol{y}_\dagger = f^*(\boldsymbol{S}_\dagger)$. According to the Riesz–Fréchet representation theorem (Lax, 2014; Schölkopf & Smola, 2002; Zhang et al., 2023b), the evaluation functional is defined as follows:

**Definition 1.** *Let $\mathcal{H}$ denote a reproducing kernel Hilbert space[2] equipped with a positive definite sequence kernel $K_{\boldsymbol{S}} \in \mathcal{H}$, where $\boldsymbol{S} \in \mathcal{S}$. The evaluation functional $E_{\boldsymbol{S}}(\cdot) : \mathcal{H} \mapsto \mathbb{R}$ is defined by the reproducing property as*

$$E_{\boldsymbol{S}}(f) = \langle f, K_{\boldsymbol{S}}(\cdot)\rangle_{\mathcal{H}} = f(\boldsymbol{S}), \quad f \in \mathcal{H} . \tag{1}$$

Furthermore, for a functional $F : \mathcal{H} \mapsto \mathbb{R}$, the Fréchet derivative (Coleman, 2012; Liu, 2017; Zhang et al., 2023b) of $F$ is defined as:

**Definition 2.** *(Fréchet derivative in RKHS) The Fréchet derivative of a functional $F : \mathcal{H} \mapsto \mathbb{R}$ at a point $f \in \mathcal{H}$, represented as $\nabla_f F(f)$, is defined implicitly by $F(f + \epsilon g) = F(f) + \langle\nabla_f F(f), \epsilon g\rangle_{\mathcal{H}} + o(\epsilon)$ for any $g \in \mathcal{H}$ and $\epsilon \in \mathbb{R}$. This derivative itself is a function in $\mathcal{H}$.*

**Attention learners**, referring to neural networks that incorporate attention mechanisms, are designed to learn the implicit mapping between input sequences and their associated properties (Vaswani et al., 2017). Specifically, the attention consists of three components: the query matrix $\mathcal{Q}(\boldsymbol{S}) \coloneqq \boldsymbol{S}\boldsymbol{W}^Q$, the key matrix $\mathcal{K}(\boldsymbol{S}) \coloneqq \boldsymbol{S}\boldsymbol{W}^K$, and the value matrix $\mathcal{V}(\boldsymbol{S}) \coloneqq \boldsymbol{S}\boldsymbol{W}^V$, where the query and key weight matrices $\boldsymbol{W}^Q$ and $\boldsymbol{W}^K$ are of size $d \times p$, and the value weight matrix $\boldsymbol{W}^V$ is of size $d \times v$. For simplicity, this paper primarily focuses on a single-layer, single-head self-attention neural network[3] (Mahankali et al., 2024; Makkuva et al., 2025), which can be expressed as

$$f_\theta(\boldsymbol{S}) = \text{softmax}\left(\frac{\mathcal{Q}(\boldsymbol{S})\mathcal{K}(\boldsymbol{S})^\top}{\sqrt{d}}\right)\mathcal{V}(\boldsymbol{S}), \tag{2}$$

where $\text{softmax}(\cdot)$ is applied row-wise.

**Nonparametric teaching** is formulated as a functional minimization over a teaching set, denoted as $\mathcal{D} = \{(\boldsymbol{x}^1, y^1), \ldots (\boldsymbol{x}^T, y^T)\}$, where each input $\boldsymbol{x} \in \mathbb{R}^d$ represents independent feature vectors, without considering the sequence (Zhang et al., 2023b). The collection of all possible teaching sets is represented by $\mathbb{D}$:

$$\mathcal{D}^* = \underset{\mathcal{D} \in \mathbb{D}}{\arg\min}\, \mathcal{M}(\hat{f}, f^*) + \lambda \cdot \text{card}(\mathcal{D}) \qquad \text{s.t.} \quad \hat{f} \coloneqq \mathcal{A}(\mathcal{D}) . \tag{3}$$

This formulation involves three key components: $\mathcal{M}$ which measures the discrepancy between $\hat{f}$ and $f^*$ (*e.g.*, $L_2$ distance in RKHS $\mathcal{M}(\hat{f}^*, f^*) = \|\hat{f}^* - f^*\|_{\mathcal{H}}$); $\text{card}(\cdot)$, representing the cardinality (or size) of the teaching set $\mathcal{D}$, controlled by a regularization constant $\lambda > 0$; and $\mathcal{A}(\mathcal{D})$, which denotes the learning algorithm employed by the learners, typically based on empirical risk minimization:

$$\mathcal{A}(\mathcal{D}) \coloneqq \underset{f \in \mathcal{H}}{\arg\min}\, \frac{1}{\text{card}(\mathcal{D})} \sum_{(\boldsymbol{x}^t, y^t) \in \mathcal{D}} \mathcal{L}\big(f(\boldsymbol{x}^t), y^t\big). \tag{4}$$

with a convex loss $\mathcal{L}$ (w.r.t. $f$), which is optimized using functional gradient descent:[4]

$$f^{t+1} \leftarrow f^t - \eta \underbrace{E_{\boldsymbol{x}}\left(\frac{\partial \mathcal{L}(f^*, f^t)}{\partial f^t}\right) \cdot K_{\boldsymbol{x}}}_{\coloneqq \mathcal{G}(\mathcal{L}, f^*; f^t, \boldsymbol{x}),\, \text{Functional Gradient}} , \tag{5}$$

where $t = 0, 1, \ldots, T$ is the iteration index, $\eta > 0$ is the learning rate, and $E_{\boldsymbol{x}}(f) = f(\boldsymbol{x})$ denotes the evaluation functional.

---

[2]In nonparametric teaching, the extension from scalar-valued to vector-valued functions, relating to element-level properties, is a well-established generalization in Zhang et al., 2023a.

[3]This can be directly extended to other attention learners, including those with multi-head attention or different types of attention mechanisms (Dong et al., 2021; Kajitsuka & Sato, 2024).

[4]The functional gradient is obtained by applying the functional chain rule (Lemma 5) and the gradient of an evaluation functional (Lemma 6), both of which are detailed in Appendix A.2.

## 4 ATTENT

We begin by investigating the role of attention in parameter-based gradient descent. Then, by translating the evolution of an ANN—driven by importance-adaptive updates in parameter space—into function space, we show that the evolution of the ANN under parameter gradient descent is consistent with that under functional gradient descent. Lastly, we present the greedy AtteNT algorithm, which effectively selects sequences with steeper gradients to enhance the learning efficiency of the ANN.

### 4.1 IMPORTANCE-ADAPTIVE UPDATE IN THE PARAMETER SPACE

Let the column vector $\theta \in \mathbb{R}^m$ denote all trainable weights in a flattened format, with $m$ representing the total number of parameters in the ANN. Figure 1 illustrates the workflow of the ANN. Given a training set of size $N$, $\{(\boldsymbol{S}_i, \boldsymbol{y}_i)|\boldsymbol{S}_i \in \mathcal{S}, \boldsymbol{y}_i \in \mathcal{Y}\}_N$, the parameters are updated via gradient descent, as shown below:[5]

$$\theta^{t+1} \leftarrow \theta^t - \frac{\eta}{NS} \sum_{i=1}^{N} \sum_{j=1}^{S} \nabla_\theta \mathcal{L}(f_{\theta^t}(\boldsymbol{S}_i)_{(j,:)}, \boldsymbol{y}_{i(j,:)}), \tag{6}$$

where $f_{\theta^t}(\boldsymbol{S}_i)_{(j,:)}$ refers to the $j$-th row of the output $f_{\theta^t}(\boldsymbol{S}_i)$, corresponding to the $j$-th element of the input sequence, and $\boldsymbol{y}_{i(j,:)}$ denotes its associated property value. Since the learning rate $\eta$ is small enough, the updates remain minimal over multiple iterations, allowing them to be treated as a time derivative and thus expressed as a differential equation (Jacot et al., 2018; Yang, 2019; Hron et al., 2020):

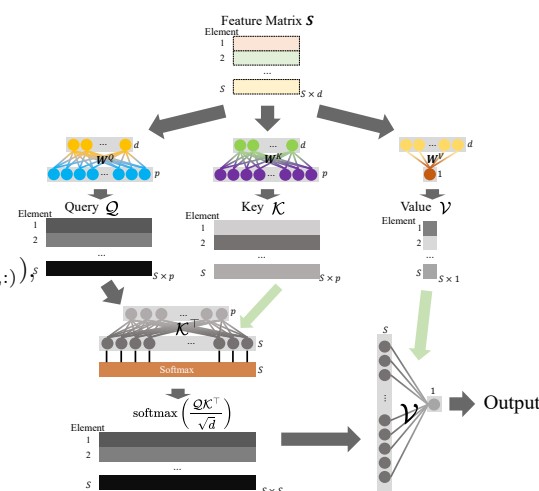

Figure 1: An illustration of the workflow for an attention neural network with an input sequence $\boldsymbol{S}$.

$$\frac{\partial \theta^t}{\partial t} = -\frac{\eta}{NS} \left[ \frac{\partial \mathcal{L}(f_{\theta^t}(\boldsymbol{S}_1), \boldsymbol{y}_1)}{\partial f_{\theta^t}(\boldsymbol{S}_1)}, \ldots, \frac{\partial \mathcal{L}(f_{\theta^t}(\boldsymbol{S}_N), \boldsymbol{y}_N)}{\partial f_{\theta^t}(\boldsymbol{S}_N)} \right] \cdot \left[ \frac{\partial f_{\theta^t}(\boldsymbol{S}_i)}{\partial \theta^t} \right]_N. \tag{7}$$

The term $\frac{\partial f_\theta(\boldsymbol{S})}{\partial \theta}$ (with the indexes $i$ and $t$ omitted for simplicity), which defines the direction for parameter updates, can be more explicitly written as

$$\frac{\partial f_\theta(\boldsymbol{S})}{\partial \theta} = \left[ \underbrace{\frac{\partial f_\theta(\boldsymbol{S})}{\partial \boldsymbol{W}_{(:,1)}^V}, \ldots, \frac{\partial f_\theta(\boldsymbol{S})}{\partial \boldsymbol{W}_{(:,v)}^V}}_{\text{w.r.t. the value weight matrix}}, \underbrace{\frac{\partial f_\theta(\boldsymbol{S})}{\partial \boldsymbol{W}_{(:,1)}^Q}, \ldots, \frac{\partial f_\theta(\boldsymbol{S})}{\partial \boldsymbol{W}_{(:,p)}^Q}}_{\text{w.r.t. the query weight matrix}}, \underbrace{\frac{\partial f_\theta(\boldsymbol{S})}{\partial \boldsymbol{W}_{(:,1)}^K}, \ldots, \frac{\partial f_\theta(\boldsymbol{S})}{\partial \boldsymbol{W}_{(:,p)}^K}}_{\text{w.r.t. the key weight matrix}} \right]. \tag{8}$$

Here, each term represents the derivative of the output $f_\theta(\boldsymbol{S})$ w.r.t. the weight column vectors. Unlike derivatives for multilayer perceptron-based learners, where the input is used only once, *i.e.*, the derivative depends on a single use of the input, the attention mechanism invokes the input three times at $\mathcal{Q}$, $\mathcal{K}$, and $\mathcal{V}$ separately, as depicted in Figure 1. To clearly demonstrate, in an analytical and explicit manner, how these three invocations allow attention to adaptively assign varying different importance to each element in an input sequence within parameter space, we present an example involving the derivative of an ANN with $v = 1$, meaning that each component of the output $f_{\theta^t}(\boldsymbol{S})_{(j,:)}$ is a scalar:

$$\frac{\partial f_\theta(\boldsymbol{S})}{\partial \theta} = \left[ \frac{\partial f_\theta(\boldsymbol{S})}{\partial \boldsymbol{W}^V}, \frac{\partial f_\theta(\boldsymbol{S})}{\partial \boldsymbol{W}_{(:,1)}^Q}, \ldots, \frac{\partial f_\theta(\boldsymbol{S})}{\partial \boldsymbol{W}_{(:,p)}^Q}, \frac{\partial f_\theta(\boldsymbol{S})}{\partial \boldsymbol{W}_{(:,1)}^K}, \ldots, \frac{\partial f_\theta(\boldsymbol{S})}{\partial \boldsymbol{W}_{(:,p)}^K} \right], \tag{9}$$

---

[5]Training sequences generally have the same length, corresponding to the maximum length, which is ensured by padding or truncating (Yu et al., 2023b; Ding et al., 2024). Therefore, this paper focuses on sequences of the same length unless noted otherwise. Results for varying lengths can be directly obtained.

where the term $\frac{\partial f_\theta(\boldsymbol{S})}{\partial \boldsymbol{W}^V}$ has a shape of $S \times d$, and is given by

$$\frac{\partial f_\theta(\boldsymbol{S})}{\partial \boldsymbol{W}^V} = \left[ \frac{\exp\left(\mathcal{Q}_{(i,:)}\mathcal{K}^\top/\sqrt{d}\right)}{\mathbf{1}^\top \exp\left(\mathcal{Q}_{(i,:)}\mathcal{K}^\top/\sqrt{d}\right)} \right]_S \boldsymbol{S}, \tag{10}$$

where $\exp(\cdot)$ denotes the element-wise exponential operator. For simplicity, we omit the arguments of $\mathcal{Q}, \mathcal{K}, \mathcal{V}$. For $i \in \mathbb{N}_p$, the term $\frac{\partial f_\theta(\boldsymbol{S})}{\partial \boldsymbol{W}^Q_{(:,i)}}$ and $\frac{\partial f_\theta(\boldsymbol{S})}{\partial \boldsymbol{W}^K_{(:,i)}}$ are

$$\frac{\partial f_\theta(\boldsymbol{S})}{\partial \boldsymbol{W}^Q_{(:,i)}} = \left[ d^{-1/2} \underbrace{\boldsymbol{S}_{(j,:)}}_{1\times d} \cdot \underbrace{\left( \overbrace{\mathcal{K}_{(:,i)}^\top}^{1\times S} \overbrace{\text{diag}\left(\text{softmax}(\mathcal{Q}_{(j,i)}\mathcal{K}_{(:,i)}^\top/\sqrt{d})\right)}^{S\times S} \overbrace{\mathcal{V}}^{S\times 1} - \overbrace{\mathcal{K}_{(:,i)}^\top}^{1\times S} \overbrace{\left(\text{softmax}(\mathcal{Q}_{(j,i)}\mathcal{K}_{(:,i)}^\top/\sqrt{d})\right)^\top \text{softmax}(\mathcal{Q}_{(j,i)}\mathcal{K}_{(:,i)}^\top/\sqrt{d})}^{S\times S} \overbrace{\mathcal{V}}^{S\times 1} \right)}_{:=\omega_j, 1\times 1} \right]_{S\times d}, \tag{11}$$

$$\frac{\partial f_\theta(\boldsymbol{S})}{\partial \boldsymbol{W}^K_{(:,i)}} = \left[ d^{-1/2} \underbrace{\boldsymbol{S}_{(j,:)}}_{1\times d} \cdot \underbrace{\left( \overbrace{\mathcal{Q}_{(:,i)}^\top}^{1\times S} \overbrace{\text{diag}\left(\text{softmax}(\mathcal{Q}_{(j,i)}\mathcal{K}_{(:,i)}^\top/\sqrt{d})\right)}^{S\times S} \overbrace{\mathcal{V}}^{S\times 1} - \overbrace{\mathcal{Q}_{(:,i)}^\top}^{1\times S} \overbrace{\left(\text{softmax}(\mathcal{Q}_{(j,i)}\mathcal{K}_{(:,i)}^\top/\sqrt{d})\right)^\top \text{softmax}(\mathcal{Q}_{(j,i)}\mathcal{K}_{(:,i)}^\top/\sqrt{d})}^{S\times S} \overbrace{\mathcal{V}}^{S\times 1} \right)}_{1\times 1} \right]_{S\times d}. \tag{12}$$

The derivation is provided in Appendix A.3. For the sake of brevity, we focus on the query gradient, with similar results holding for the key and value gradients. As a result of invoking the input three times, Equation 11 reveals that the ANN gradient depends not only on the features of the sequence elements, *i.e.*, $\boldsymbol{S}_{(j,:)}$, but also on a scalar $\omega_j$ that is specific to each element.

Specifically, Equation 11 explicitly shows that the gradient row order follows the order of elements in the input sequence, meaning the gradient is equivariant w.r.t. reordering the elements. This is in contrast to the gradient in recurrent neural networks (Elman, 1990; Jordan, 1997), where the order of the elements determines the power of the recurrent weights. Moreover, this gradient property is derived during the training stage, yet, interestingly, it aligns with the permutation invariance property of self-attention during inference (Lee et al., 2019).

The scalar $\omega_j$ in Equation 11 is computed from $\mathcal{Q}, \mathcal{K},$ and $\mathcal{V}$, which reflects the three invocations of input by the attention. It is clear that it is closely associated with the $j$-th element, meaning it is element-specific. This scalar is the importance value that attention assigns to each element, leading to an importance-adaptive update in the parameter space. When all importance values are set to 1, the gradient of the ANN reduces to the derivative of a multilayer perceptron without nonlinear activations and with batch input. Additionally, the explicit expressions in Equations 7, 11, and 12 show that the ANN gradient does not depend on the input sequence length (*i.e.*, the number of elements), as this is averaged out. Instead, it depends on the feature dimension. In other words, the parameter gradient remains unchanged even if the input sequence length $S$ is scaled.

## 4.2 THE FUNCTIONAL EVOLUTION OF ANN

The importance-adaptive update in the parameter space drives the functional evolution of $f_\theta \in \mathcal{H}$. This variation in $f_\theta$, reflecting how $f_\theta$ responds to updates in $\theta$, can be derived using Taylor's theorem as follows:

$$f(\theta^{t+1}) - f(\theta^t) = \langle \nabla_\theta f(\theta^t), \theta^{t+1} - \theta^t \rangle + o(\theta^{t+1} - \theta^t), \tag{13}$$

where $f(\theta^\dagger) \equiv f_{\theta^\dagger}$. In a manner analogous to the transformation of parameter updates into their differential form, this can also be expressed in a differential form (Zhang et al., 2024a):

$$\frac{\partial f_{\theta^t}}{\partial t} = \underbrace{\left\langle \frac{\partial f(\theta^t)}{\partial \theta^t}, \frac{\partial \theta^t}{\partial t} \right\rangle}_{(*)} + o\left(\frac{\partial \theta^t}{\partial t}\right). \tag{14}$$

By substituting the specific parameter updates, *i.e.*, Equation 7, into the first-order approximation term $(*)$ of this variation, we obtain

$$\frac{\partial f_{\theta^t}}{\partial t} = -\frac{\eta}{NS} \left[ \frac{\partial \mathcal{L}(f_{\theta^t}(\boldsymbol{S}_1), \boldsymbol{y}_1)}{\partial f_{\theta^t}(\boldsymbol{S}_1)}, \dots, \frac{\partial \mathcal{L}(f_{\theta^t}(\boldsymbol{S}_N), \boldsymbol{y}_N)}{\partial f_{\theta^t}(\boldsymbol{S}_N)} \right] \cdot [K_{\theta^t}(\boldsymbol{S}_i, \cdot)]_N + o\left(\frac{\partial \theta^t}{\partial t}\right), \tag{15}$$

where the symmetric and positive definite $K_{\theta^t}(\boldsymbol{S}_i, \cdot) := \left\langle \frac{\partial f_{\theta^t}(\boldsymbol{S}_i)}{\partial \theta^t}, \frac{\partial f_{\theta^t}(\cdot)}{\partial \theta^t} \right\rangle$ (for detailed derivations and further discussion, see Appendix A.4). Due to the inclusion of nonlinear activation functions in

$f(\theta)$, the nonlinearity of $f(\theta)$ with respect to $\theta$ results in the remainder $o(\theta^{t+1} - \theta^t)$ being nonzero. In a subtle difference, Jacot et al., 2018; Yang, 2019; Hron et al., 2020 apply the chain rule directly, giving less focus to the convexity of $\mathcal{L}$ with respect to $\theta$. As a result, the first-order approximation is derived as the variation, with $K_\theta$ being referred to as the Attention Neural Tangent Kernel (ANTK). It has been demonstrated that the ANTK remains constant during training when the ANN width, *i.e.*, $d$, is assumed to be infinite (Hron et al., 2020). However, in practical applications, the ANN width does not need to be infinitely large, prompting us to explore the dynamic ANTK (an example of how the ANTK is computed can be found in Figure 3 in Appendix A.4).

Consider characterizing the variation of $f_\theta \in \mathcal{H}$ from a high-level, functional viewpoint (Zhang et al., 2024a; 2025). Using functional gradient descent, it can be written as

$$\frac{\partial f_{\theta^t}}{\partial t} = -\eta \mathcal{G}(\mathcal{L}, f^*; f_{\theta^t}, \{\boldsymbol{S}_i\}_N), \tag{16}$$

where the functional gradient is expressed as

$$\mathcal{G}(\mathcal{L}, f^*; f_{\theta^t}, \{\boldsymbol{S}_i\}_N) = \frac{1}{NS} \left[ \frac{\partial \mathcal{L}(f_{\theta^t}(\boldsymbol{S}_1), \boldsymbol{y}_1)}{\partial f_{\theta^t}(\boldsymbol{S}_1)}, \ldots, \frac{\partial \mathcal{L}(f_{\theta^t}(\boldsymbol{S}_N), \boldsymbol{y}_N)}{\partial f_{\theta^t}(\boldsymbol{S}_N)} \right] [K(\boldsymbol{S}_i, \cdot)]_N . \tag{17}$$

The asymptotic relationship between ANTK and the importance-adaptive canonical kernel (Cancedda et al., 2003; Király & Oberhauser, 2019; Zhang et al., 2024a) in the context of functional gradient is presented in Theorem 3 below, with the proof provided in Appendix B.1.

**Theorem 3.** *Given a convex loss $\mathcal{L}$ and a training set $\{(\boldsymbol{S}_i, \boldsymbol{y}_i) | \boldsymbol{S}_i \in \mathcal{S}, \boldsymbol{y}_i \in \mathcal{Y}\}_N$, the dynamic ANTK, which is derived from performing gradient descent on the parameters of an ANN, converges pointwise to the importance-adaptive canonical kernel in the dual functional gradient with respect to the input sequences. Specifically, it holds that*

$$\lim_{t \to \infty} K_{\theta^t}(\boldsymbol{S}_i, \cdot) = K(\boldsymbol{S}_i, \cdot), \quad \forall i \in \mathbb{N}_N . \tag{18}$$

This suggests that ANTK, which includes adaptive importance information, serves as a dynamic substitute for the importance-adaptive canonical kernel in functional gradient descent with sequence inputs, aligning the ANN evolution through parameter gradient descent with that in functional gradient descent (Kuk, 1995; Hron et al., 2020; Geifman et al., 2020). This functional insight bridges the teaching of attention learners (*i.e.*, ANNs) with that of importance-adaptive nonparametric learners, while also facilitating further analysis (*e.g.*, a convex functional $\mathcal{L}$ retains its convexity with respect to $f_\theta$ from a functional perspective, but is typically nonconvex when considering $\theta$). By utilizing the functional insight and applying the canonical kernel (Dou & Liang, 2021) instead of ANTK (which should be considered *alongside the remainder*), it facilitates deriving sufficient reduction concerning $\mathcal{L}$ in Proposition 4, with the proof deferred to Appendix B.2.

**Proposition 4.** *(Sufficient Loss Reduction) Let the convex loss $\mathcal{L}$ be Lipschitz smooth with a constant $\tau > 0$, and suppose the importance-adaptive canonical kernel is bounded above by a constant $\gamma > 0$. If the learning rate $\eta$ satisfies $\eta \leq 1/(2\tau\gamma)$, then a sufficient reduction in $\mathcal{L}$ is guaranteed, as demonstrated by*

$$\frac{\partial \mathcal{L}}{\partial t} \leq -\frac{\eta\gamma}{2} \left( \frac{1}{NS} \sum_{i=1}^{N} \sum_{j=1}^{S} \frac{\partial \mathcal{L}\left( f_{\theta^t}(\boldsymbol{S}_i)_{(j,:)}, \boldsymbol{y}_{i(j,:)} \right)}{\partial f_{\theta^t}(\boldsymbol{S}_i)_{(j,:)}} \right)^2 . \tag{19}$$

This indicates that the variation of $\mathcal{L}$ over time is capped by a negative value, meaning it decreases by at least the magnitude of this upper bound as time progresses, ensuring convergence.

### 4.3 THE ATTENT ALGORITHM

Building on the understanding of how attention adaptively assigns varying importance in parameter-based gradient descent, as well as the consistency between teaching an ANN and a nonparametric learner, we introduce the AtteNT algorithm. This algorithm is designed to amplify the steepness of the gradients, thereby improving the learning efficiency of the ANN. By considering the gradient as the sum of projections of $\frac{\partial \mathcal{L}(f_\theta, f^*)}{\partial f_\theta}$ onto the basis $\{K(\boldsymbol{S}_i, \cdot)\}_N$, the gradient can be increased simply

by maximizing the projection $\frac{\partial \mathcal{L}(f_\theta(\boldsymbol{S}_i), \boldsymbol{y}_i)}{\partial f_\theta(\boldsymbol{S}_i)}$, thus eliminating the need to compute the norm of the basis $\|K(\boldsymbol{S}_i, \cdot)\|_{\mathcal{H}}$ (Wright, 2015; Zhang et al., 2024a). This suggests that selecting sequences that either maximize $\left\|\frac{\partial \mathcal{L}(f_\theta(\boldsymbol{S}_i), \boldsymbol{y}_i)}{\partial f_\theta(\boldsymbol{S}_i)}\right\|_2$ or correspond to the larger components of $\frac{\partial \mathcal{L}(f_\theta, f^*)}{\partial f_\theta}$ can effectively amplify the gradient, indicating that

$$\{\boldsymbol{S}_i\}_m{}^* = \underset{\{\boldsymbol{S}_i\}_m \subseteq \{\boldsymbol{S}_i\}_N}{\arg \max} \left\| \left[ \frac{\partial \mathcal{L}(f_\theta(\boldsymbol{S}_i), \boldsymbol{y}_i)}{\partial f_\theta(\boldsymbol{S}_i)} \right]_m \right\|_{\mathcal{F}}, \tag{20}$$

with Frobenius norm $\|\cdot\|_{\mathcal{F}}$. From a functional viewpoint, for a convex loss functional $\mathcal{L}$, the norm of its partial derivative w.r.t. $f_\theta$, denoted as $\|\frac{\partial \mathcal{L}(f_\theta)}{\partial f_\theta}\|_{\mathcal{H}}$, is positively correlated with $\|f_\theta - f^*\|_{\mathcal{H}}$. As $f_\theta$ gets closer to $f^*$, the value of $\|\frac{\partial \mathcal{L}(f_\theta)}{\partial f_\theta}\|_{\mathcal{H}}$ decreases (Boyd & Vandenberghe, 2004; Coleman, 2012). This relationship becomes especially prominent when $\mathcal{L}$ is strongly convex with a larger convexity constant (Kakade & Tewari, 2008; Arjevani et al., 2016). Building on these insights, the AtteNT algorithm selects sequences by

$$\{\boldsymbol{S}_i\}_m{}^* = \underset{\{\boldsymbol{S}_i\}_m \subseteq \{\boldsymbol{S}_i\}_N}{\arg \max} \|[f_\theta(\boldsymbol{S}_i) - f^*(\boldsymbol{S}_i)]_m\|_{\mathcal{F}}. \tag{21}$$

The pseudocode is provided in Algorithm 1.

## 5 EXPERIMENTS AND RESULTS

To demonstrate the broad effectiveness of the AtteNT Algorithm, we conducted extensive experiments across diverse domains. Our evaluation covered large language models and computer vision models. In addition, we validated performance under multiple training paradigms, including training from scratch, and fine-tuning, consistently achieving strong results.

**LLM Scenario.** We evaluate AtteNT algorithms across a diverse set of natural language generation (NLG) tasks. Specifically, we fine-tune LLaMA 2-7B (Touvron et al., 2023), Mistral-7B (Jiang et al., 2023), and Gemma-7B (Team et al., 2024) on the MetaMathQA dataset (Yu et al., 2023a) to benchmark their mathematical reasoning capabilities on GSM8K (Cobbe et al., 2021) and MATH (Hendrycks et al., 2021). To assess coding proficiency, we further fine-tune these models on CodeFeedback (Zheng et al., 2024b) and evaluate on HumanEval (Chen et al., 2021a) and MBPP (Austin et al., 2021). For conversational ability, we train on WizardLM-Evol-Instruct (Xu et al., 2023) and evaluate on MT-Bench (Zheng et al., 2024a). All experiments are conducted on standardized subsets to ensure comparable training efficiency and are trained for five epochs.

As shown in Table 1, AtteNT consistently outperforms standard fine-tuning across all evaluated models and tasks while reducing computational overhead. Specifically, fine-tuning LLaMA, Mistral, and Gemma with AtteNT yields accuracy gains of 1.39, 2.14, and 2.42 on GSM8K, and 1.59, 2.89, and 0.76 on MATH. On coding benchmarks, AtteNT improves performance by 3.66%, 3.25%, and 0.29% on HumanEval, and by 2.08%, 3.25%, and 3.31% on MBPP. We further report average fine-tuning time per model under identical data volumes and epoch settings. Since runtime variation arises primarily from AtteNT's adaptive data selection, the observed results highlight its efficiency: on average, AtteNT reduces training time by 12.78%, underscoring its advantage in both performance and resource savings.

Table 1: AtteNT on NLG tasks. The results are averaged over three runs, with standard deviations included. The GSM8K and MATH datasets share a math fine-tuned model, while HumanEval and MBPP use a code fine-tuned model. MT-Bench utilizes a conversation fine-tuned model. The "Avg. time" represents the average fine-tuning time for the three models.

| Model | AtteNT | Avg. Time(↓) | GSM8K(↑) | MATH(↑) | HumanEval(↑) | MBPP(↑) | MT-Bench(↑) |
|---|---|---|---|---|---|---|---|
| LLaMA 2-7B | w/o | 246±1m | 42.96±0.12 | 5.06±0.16 | 18.35±0.31 | 35.65±0.25 | **4.58±0.01** |
| | w | **213±2m** | **43.45±0.55** | **6.48±0.24** | **21.80±0.38** | **37.61±0.42** | 4.49±0.02 |
| Mistral-7B | w/o | 204±2m | 69.13±0.22 | 20.06±0.20 | 43.42±0.14 | 58.52±0.13 | 5.03±0.05 |
| | w | **180±2m** | **71.26±0.23** | **23.12±0.44** | **46.55±0.25** | **61.74±0.54** | **5.32±0.04** |
| Gemma-7B | w/o | 228±2m | 75.23±0.45 | 30.52±0.48 | 53.83±0.27 | 65.69±0.29 | 5.42±0.04 |
| | w | **201±2m** | **77.74±0.32** | **31.40±0.36** | **54.26±0.28** | **66.28±0.46** | **5.44±0.08** |

Table 2: AtteNT across various CV downstream tasks. ImageNetS50 uses 50 categories from ImageNet for classification, evaluated by accuracy. NYUv2(S) is a semantic segmentation task with mIoU as the metric. NYUv2(D) involves depth estimation, evaluated using the $\delta_1$ metric, which measures the percentage of pixels with an error ratio below 1.25 (Doersch & Zisserman, 2017).

| Model | AtteNT | Pretraining Time($\downarrow$) | ImageNetS50($\uparrow$) | NYUv2(S)($\uparrow$) | NYUv2(D)($\uparrow$) |
|---|---|---|---|---|---|
| Multi-Modal MAE | w/o | 1234m | 92.2 | 51.9 | 52.1 |
|  | w | **980m**(-20.58%) | **92.3** | **52.6** | **57.2** |

Table 3: Ablation study of AtteNT pre-training configurations. Ratio controls how the fraction of selected samples increases over epochs. Interval denotes how often the subset is re-sampled. Selection specifies the sampling strategy: Random (no difficulty prior), Hard (selects only difficult samples), and Soft (Gumbel-Top-k difficulty-aware sampling). The configuration (Incremental, Incremental, Soft) in the red color row is adopted as our final AtteNT setting, as it simultaneously reduces pre-training time and improves performance on all downstream tasks.

| Pre-training | | | | Downstream | | |
|---|---|---|---|---|---|---|
| Ratio | Interval | Selection | Training time($\downarrow$) | ImageNetS50($\uparrow$) | NYUv2(S)($\uparrow$) | NYUv2(D)($\uparrow$) |
| - | - | - | 1234m | 92.2 | 51.9 | 52.1 |
| Cosine | Incremental | Random | 966m | 88.6 | 45.3 | 49.6 |
| Cosine | Incremental | Soft | 995m | 92.1 | 52.2 | 58.8 |
| Cosine | Fixed | Soft | 1301m | **93.2** | 53.6 | 61.4 |
| Incremental | Incremental | Soft | 980m | 92.3 | 52.6 | 57.2 |
| Incremental | Fixed | Soft | 1319m | 92.4 | **53.7** | **62.1** |
| Cosine | Incremental | Hard | 972m | 91.8 | 49.5 | 57.3 |
| Cosine | Fixed | Hard | 1285m | 92.1 | 53.0 | 60.8 |
| Incremental | Incremental | Hard | **963m** | 91.4 | 48.4 | 57.2 |
| Incremental | Fixed | Hard | 1302m | 92.5 | 52.7 | 59.5 |

**CV Scenario.** The Multi-Modal MAE (Bachmann et al., 2022) is designed to address a diverse range of downstream tasks by employing three specialized encoders, each dedicated to processing a distinct image modality. During pre-training, we explore various selection strategies, including different ratios and intervals, to optimize model configuration. The pretraining process is conducted over 800 epochs.

For unsupervised pre-training, we utilize ImageNetS50 (Gao et al., 2021) to evaluate the effectiveness of the AttneNT method in enhancing the performance of downstream tasks under suboptimal conditions. Classification performance is assessed using the validation subset of the original dataset, while semantic segmentation and depth estimation tasks are fine-tuned and evaluated on the NYUv2 dataset (Silberman et al., 2012). Given the absence of a large multi-task dataset with aligned task-specific images (Doersch & Zisserman, 2017; Bachmann et al., 2022; Wang et al., 2023), we generate pseudo-labels for ImageNetS50 using Mask2Former (Cheng et al., 2022).

As shown in Table 2, the AteNT strategy results in a significant reduction in training time, saving 20.58% during long-duration training from scratch. Additionally, it consistently improves performance across a wide range of downstream tasks. Notably, the depth estimation task exhibits the largest gain, achieving a 5.1% improvement. We attribute this improvement to the nature of the depth estimation task, which is independent of image type, thus preventing any disruption in data distribution during the selection process. Our experiments demonstrate the efficacy of AtteNT within the ViT architecture.

The practical performance gains stem directly from the curriculum effect induced by nonparametric teaching (Bengio et al., 2009; Wang et al., 2021b; Zhang et al., 2023b; 2025), which greedily selects the examples that most advance the learner. This naturally creates a curriculum that focuses training on informative, high-gradient examples and avoids gradient dilution from already-mastered ones.

We further present the ablation study results for AttneNT, focusing on the effects of varying data selection strategies and their impact on downstream tasks. Specifically, we investigate the influence of dynamic changes in data selection ratios and step sizes, following the strategy proposed in (Zhang et al., 2023b). Additionally, we examine how different selection strategies affect the performance

of downstream tasks. The Random strategy involves selecting data without any predefined criteria, while the Hard strategy entails deterministic data selection. The Soft strategy, on the other hand, uses probability-based data selection, derived from loss scores. To implement this, we apply the Gumbel-Top-k selection algorithm (Kool et al., 2019) for sampling without replacement. Our results show that the Soft selection strategy achieves the best performance in downstream tasks, significantly improving the model's robustness during training. A more detailed study of the sample ratio can be found in Appendix D.1, and additional comparison results are provided in Appendix D.2.

## 6 CONCLUDING REMARKS AND FUTURE WORK

This paper introduces AtteNT, a novel paradigm that enhances the learning efficiency of attention learners (*i.e.*, ANNs) through nonparametric teaching theory. Specifically, AtteNT reduces the wallclock time required to learn the implicit mapping from sequences to relevant properties by 13.01% to 20.58% while consistently preserving and often enhancing the performance across a diverse set of downstream tasks. Moreover, AtteNT establishes a theoretical connection between the evolution of an ANN via parameter-based gradient descent and that of a function using functional gradient descent in nonparametric teaching. This connection between nonparametric teaching theory and ANN training expands the potential applications of nonparametric teaching in contexts that involve attention mechanisms.

In future work, it would be interesting to explore other variations of AtteNT for different attention learners, such as graph attention networks (Veličković et al., 2018). Additionally, investigating its robustness under real-world label noise, building upon recent noise-robust advancements (Wei et al., 2024; Hu et al., 2024), could yield crucial improvements. Another promising direction is to examine the practical applications of AtteNT in improving the efficiency of data-driven methods (Henaff, 2020; Touvron et al., 2021; Müller et al., 2022) for attention-related tasks, especially in areas like world models.

## REPRODUCIBILITY STATEMENT

We have taken substantial steps to promote the reproducibility of our research. Appendix A offers a comprehensive overview of the notation, theoretical background, and key algorithm. All proofs for theorems and propositions can be found in Appendix B. Meanwhile, Appendix C provides a comprehensive description of the experimental setup, including training configurations, hyperparameter choices, algorithmic details, and dataset preprocessing procedures. Codes are available at the following link: LINK.

## ACKNOWLEDGMENTS

This work was supported in part by the Theme-based Research Scheme (TRS) project T45-701/22-R of the Research Grants Council of Hong Kong, and in part by the AVNET-HKU Emerging Microelectronics and Ubiquitous Systems (EMUS) Lab. For computer time, this research used the resources of the Supercomputing Laboratory at KAUST.

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

# Appendix

# A    ADDITIONAL DISCUSSIONS

## A.1    NOTATION OVERVIEW

Table 4: Summary of Key Notations.

| Notation | Description |
|---|---|
| $\boldsymbol{S}_{S \times d}$ | Matrix containing all feature vectors from the ordered sequence $(\boldsymbol{x}_1, \ldots, \boldsymbol{x}_S)$, with shape $S \times d$ |
| $[x_{s,j}]_d^\top$ | $d$-dimensional feature vector for the $s$-th element, with components $x_{s,j}$ |
| $\boldsymbol{x}$ | Short form for $[x_j]_d$ |
| $\boldsymbol{S}_{(s,:)}$ | The $s$-th row of $\boldsymbol{S}$, representing the feature vector for the $s$-th element |
| $\boldsymbol{S}_{(:,i)}$ | The $i$-th column of $\boldsymbol{S}$, which represents the $i$-th feature across all elements) |
| $\boldsymbol{e}_i$ | The $i$-th basis vector, having a value of 1 at the $i$-th position and 0 elsewhere |
| $\mathcal{S}$ | Collection of all sequences |
| $\boldsymbol{y}$ | Property associated with the sequences, which can be scalar or vector |
| $\mathcal{Y}$ | Space of sequential properties, represented as $\mathbb{R}$ or $\mathbb{R}^n$ |
| $\{a_i\}_m$ | A set containing $m$ items |
| $\mathrm{diag}(a_1, \ldots, a_m)$ | Diagonal matrix with diagonal entries $a_1, \ldots, a_m$ |
| $\mathrm{diag}(a; m)$ | Diagonal matrix with $m$ repeated entries of $a$ |
| $\mathbb{N}_S \coloneqq \{1, \ldots, S\}$ | Set of natural numbers from 1 to $S$ |
| $K(\boldsymbol{S}, \boldsymbol{S}')$ | A symmetric and positive definite sequence kernel |
| $\mathcal{H}$ | Reproducing kernel Hilbert space (RKHS) defined by $K$ |
| $f^*$ | Target mapping from $\mathcal{S}$ to $\mathcal{Y}$ |
| $\boldsymbol{y}_\dagger$ | Property $f^*(\boldsymbol{S}_\dagger)$ corresponding to the sequence $\boldsymbol{S}_\dagger$ |

## A.2    FUNCTIONAL GRADIENT

Zhang et al., 2023b;a present the chain rule for functional gradients, which is detailed in Lemma 5 (Gelfand & Silverman, 2000), and utilize the Fréchet derivative to calculate the derivative of the evaluation functional in RKHS, as shown in Lemma 6 (Coleman, 2012).

**Lemma 5.** *(Chain rule for functional gradients) For differentiable functions $G(F) : \mathbb{R} \mapsto \mathbb{R}$ that depend on functionals $F(f) : \mathcal{H} \mapsto \mathbb{R}$, the chain rule is given by*

$$\nabla_f G(F(f)) = \frac{\partial G(F(f))}{\partial F(f)} \cdot \nabla_f F(f) \,. \tag{22}$$

**Lemma 6.** *The gradient of the evaluation functional at the feature $\boldsymbol{x}$, denoted as $E_{\boldsymbol{x}}(f) = f(\boldsymbol{x}) : \mathcal{H} \to \mathbb{R}$, is given by $\nabla_f E_{\boldsymbol{x}}(f) = K(\boldsymbol{x}, \cdot)$, where $K(\boldsymbol{x}, \boldsymbol{x}') : \mathbb{R}^d \times \mathbb{R}^d \to \mathbb{R}$ represents a feature-based kernel.*

## A.3    THE DERIVATION OF IMPORTANCE-ADAPTIVE UPDATES IN THE PARAMETER SPACE.

Before providing the detailed derivation, we begin by showing visualizations of general single-head attention learners. Figure 2a depicts a multi-output self-attention learner, Figure 2b presents a multi-output masked self-attention learner, and Figure 2c illustrates a multi-output cross-attention learner. The formulations for the masked self-attention and cross-attention learners are presented in Equation 23.

$$\text{Masked Self-Attention:} \quad f_\theta(\boldsymbol{S}) = \mathrm{softmax}\left(\frac{\mathcal{Q}(\boldsymbol{S})\mathcal{K}(\boldsymbol{S})^\top}{\sqrt{d}} + \boldsymbol{M}\right) \mathcal{V}(\boldsymbol{S})$$

$$\text{Cross-Attention:} \quad f_\theta(\boldsymbol{S}, \boldsymbol{S}') = \mathrm{softmax}\left(\frac{\mathcal{Q}(\boldsymbol{S})\mathcal{K}(\boldsymbol{S}')^\top}{\sqrt{d}}\right) \mathcal{V}(\boldsymbol{S}'), \tag{23}$$

where $\boldsymbol{M} \in \mathbb{R}^{S \times S}$ is a is a strictly upper triangular matrix, with zeros on and below the diagonal and $-\infty$ in every element above the diagonal.

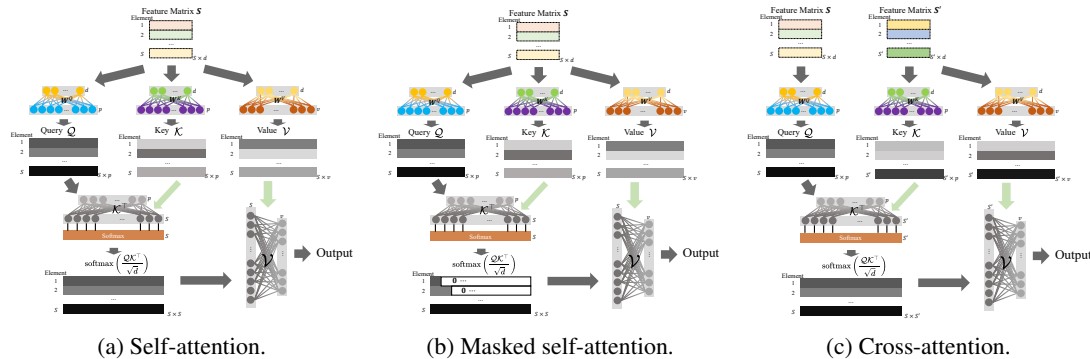

(a) Self-attention.      (b) Masked self-attention.      (c) Cross-attention.

Figure 2: An illustration of the workflow for different multi-output attention learners, with input sequence $\boldsymbol{S}$ and $\boldsymbol{S}'$ (in the case of cross-attention).

Consider the derivative of an ANN with $v = 1$, meaning that each component of the output $f_{\theta^t}(\boldsymbol{S})_{(j,:)}$ is a scalar:

$$\frac{\partial f_\theta(\boldsymbol{S})}{\partial \theta} = \left[ \frac{\partial f_\theta(\boldsymbol{S})}{\partial \boldsymbol{W}^V}, \frac{\partial f_\theta(\boldsymbol{S})}{\partial \boldsymbol{W}^Q_{(:,1)}}, \ldots, \frac{\partial f_\theta(\boldsymbol{S})}{\partial \boldsymbol{W}^Q_{(:,p)}}, \frac{\partial f_\theta(\boldsymbol{S})}{\partial \boldsymbol{W}^K_{(:,1)}}, \ldots, \frac{\partial f_\theta(\boldsymbol{S})}{\partial \boldsymbol{W}^K_{(:,p)}} \right]. \tag{24}$$

By applying the chain rule, we can compute the derivative of $f_\theta(\boldsymbol{S})$ with respect to the weight $\boldsymbol{W}^V$ in the value matrix $\mathcal{V}(\boldsymbol{S})$.

$$\begin{aligned}
\frac{\partial f_\theta(\boldsymbol{S})}{\partial \boldsymbol{W}^V} &= \frac{\partial \operatorname{softmax}\left(\frac{\mathcal{Q}(\boldsymbol{S})\mathcal{K}(\boldsymbol{S})^\top}{\sqrt{d}}\right)\mathcal{V}(\boldsymbol{S})}{\partial \boldsymbol{W}^V} \\
&= \frac{\partial \operatorname{softmax}\left(\frac{\boldsymbol{S}\boldsymbol{W}^Q\boldsymbol{W}^{K\top}\boldsymbol{S}^\top}{\sqrt{d}}\right)\boldsymbol{S}\boldsymbol{W}^V}{\partial \boldsymbol{W}^V} \\
&= \operatorname{softmax}\left(\frac{\boldsymbol{S}\boldsymbol{W}^Q\boldsymbol{W}^{K\top}\boldsymbol{S}^\top}{\sqrt{d}}\right)\boldsymbol{S} \\
&= \operatorname{softmax}\left(\frac{\mathcal{Q}\mathcal{K}^\top}{\sqrt{d}}\right)\boldsymbol{S} \\
&= \left[ \frac{\exp\left(\mathcal{Q}_{(i,:)}\mathcal{K}^\top/\sqrt{d}\right)}{\mathbf{1}^\top \exp\left(\mathcal{Q}_{(i,:)}\mathcal{K}^\top/\sqrt{d}\right)} \right]_S \boldsymbol{S},
\end{aligned} \tag{25}$$

where $\exp(\cdot)$ denotes the row-wise exponential operator. The case of $v \geq 2$ represents a multi-dimensional extension, which involves more complex notation but can be derived in a similar manner.

The derivative of $f_\theta(\boldsymbol{S})$ with respect to the query weight matrix is more intricate. For $i \in \mathbb{N}_p$,

$$
\begin{aligned}
\frac{\partial f_\theta(\boldsymbol{S})}{\partial \boldsymbol{W}_{(:,i)}^Q} &= \frac{\partial \operatorname{softmax}\left(\frac{\mathcal{Q}(\boldsymbol{S})\mathcal{K}(\boldsymbol{S})^\top}{\sqrt{d}}\right)\mathcal{V}(\boldsymbol{S})}{\partial \boldsymbol{W}_{(:,i)}^Q} \\[2mm]
&= \frac{\partial \operatorname{softmax}\left(\frac{\boldsymbol{S}\boldsymbol{W}^Q\boldsymbol{W}^{K\top}\boldsymbol{S}^\top}{\sqrt{d}}\right)\boldsymbol{S}\boldsymbol{W}^V}{\partial \boldsymbol{W}_{(:,i)}^Q} \\[2mm]
&= \frac{\partial \operatorname{softmax}\left(\frac{\boldsymbol{S}\boldsymbol{W}^Q\boldsymbol{e}_i\boldsymbol{e}_i^\top\boldsymbol{W}^{K\top}\boldsymbol{S}^\top}{\sqrt{d}}\right)\boldsymbol{S}\boldsymbol{W}^V}{\partial \boldsymbol{W}_{(:,i)}^Q} \\[2mm]
&= \begin{bmatrix} \frac{\partial \operatorname{softmax}\left(\frac{\boldsymbol{S}_{(1,:)}\boldsymbol{W}_{(:,i)}^Q\boldsymbol{W}_{(:,i)}^{K\top}\boldsymbol{S}^\top}{\sqrt{d}}\right)\boldsymbol{S}\boldsymbol{W}^V}{\partial \boldsymbol{W}_{(:,i)}^Q} \\ \cdots \\ \frac{\partial \operatorname{softmax}\left(\frac{\boldsymbol{S}_{(S,:)}\boldsymbol{W}_{(:,i)}^Q\boldsymbol{W}_{(:,i)}^{K\top}\boldsymbol{S}^\top}{\sqrt{d}}\right)\boldsymbol{S}\boldsymbol{W}^V}{\partial \boldsymbol{W}_{(:,i)}^Q} \end{bmatrix} \\[2mm]
&= \begin{bmatrix} \partial\frac{\boldsymbol{S}_{(1,:)}\boldsymbol{W}_{(:,i)}^Q\boldsymbol{W}_{(:,i)}^{K\top}\boldsymbol{S}^\top}{\sqrt{d}}\bigg/\partial\boldsymbol{W}_{(:,i)}^Q \; \partial \operatorname{softmax}\left(\frac{\boldsymbol{S}_{(1,:)}\boldsymbol{W}_{(:,i)}^Q\boldsymbol{W}_{(:,i)}^{K\top}\boldsymbol{S}^\top}{\sqrt{d}}\right)\bigg/\partial\frac{\boldsymbol{S}_{(1,:)}\boldsymbol{W}_{(:,i)}^Q\boldsymbol{W}_{(:,i)}^{K\top}\boldsymbol{S}^\top}{\sqrt{d}}\,\boldsymbol{S}\boldsymbol{W}^V \\ \cdots \\ \partial\frac{\boldsymbol{S}_{(S,:)}\boldsymbol{W}_{(:,i)}^Q\boldsymbol{W}_{(:,i)}^{K\top}\boldsymbol{S}^\top}{\sqrt{d}}\bigg/\partial\boldsymbol{W}_{(:,i)}^Q \; \partial \operatorname{softmax}\left(\frac{\boldsymbol{S}_{(S,:)}\boldsymbol{W}_{(:,i)}^Q\boldsymbol{W}_{(:,i)}^{K\top}\boldsymbol{S}^\top}{\sqrt{d}}\right)\bigg/\partial\frac{\boldsymbol{S}_{(S,:)}\boldsymbol{W}_{(:,i)}^Q\boldsymbol{W}_{(:,i)}^{K\top}\boldsymbol{S}^\top}{\sqrt{d}}\,\boldsymbol{S}\boldsymbol{W}^V \end{bmatrix}.
\end{aligned} \tag{26}
$$

Let's examine this row by row. For the $j$-th row ($j \in \mathbb{N}_S$) of Equation 26, it expressed as:

$$
\begin{aligned}
&\frac{\partial\frac{\boldsymbol{S}_{(j,:)}\boldsymbol{W}_{(:,i)}^Q\boldsymbol{W}_{(:,i)}^{K\top}\boldsymbol{S}^\top}{\sqrt{d}}}{\partial\boldsymbol{W}_{(:,i)}^Q}\; \frac{\partial \operatorname{softmax}\left(\frac{\boldsymbol{S}_{(j,:)}\boldsymbol{W}_{(:,i)}^Q\boldsymbol{W}_{(:,i)}^{K\top}\boldsymbol{S}^\top}{\sqrt{d}}\right)}{\partial\frac{\boldsymbol{S}_{(j,:)}\boldsymbol{W}_{(:,i)}^Q\boldsymbol{W}_{(:,i)}^{K\top}\boldsymbol{S}^\top}{\sqrt{d}}}\,\boldsymbol{S}\boldsymbol{W}^V \\[2mm]
&= \boldsymbol{S}_{(j,:)} \cdot \frac{1}{\sqrt{d}}\boldsymbol{W}_{(:,i)}^{K\top}\boldsymbol{S}^\top\; \frac{\partial \operatorname{softmax}\left(\frac{\boldsymbol{S}_{(j,:)}\boldsymbol{W}_{(:,i)}^Q\boldsymbol{W}_{(:,i)}^{K\top}\boldsymbol{S}^\top}{\sqrt{d}}\right)}{\partial\frac{\boldsymbol{S}_{(j,:)}\boldsymbol{W}_{(:,i)}^Q\boldsymbol{W}_{(:,i)}^{K\top}\boldsymbol{S}^\top}{\sqrt{d}}}\,\boldsymbol{S}\boldsymbol{W}^V \\[2mm]
&= \boldsymbol{S}_{(j,:)} \cdot \frac{1}{\sqrt{d}}\boldsymbol{W}_{(:,i)}^{K\top}\boldsymbol{S}^\top\; \frac{\partial\left(\frac{\exp\left(d^{-1/2}\boldsymbol{S}_{(j,:)}\boldsymbol{W}_{(:,i)}^Q\boldsymbol{W}_{(:,i)}^{K\top}\boldsymbol{S}^\top\right)}{\mathbf{1}^\top\exp\left(d^{-1/2}\boldsymbol{S}_{(j,:)}\boldsymbol{W}_{(:,i)}^Q\boldsymbol{W}_{(:,i)}^{K\top}\boldsymbol{S}^\top\right)}\right)}{\partial\, d^{-1/2}\boldsymbol{S}_{(j,:)}\boldsymbol{W}_{(:,i)}^Q\boldsymbol{W}_{(:,i)}^{K\top}\boldsymbol{S}^\top}\,\boldsymbol{S}\boldsymbol{W}^V \\[2mm]
&= \boldsymbol{S}_{(j,:)} \cdot \Bigg(\frac{1}{\sqrt{d}}\boldsymbol{W}_{(:,i)}^{K\top}\boldsymbol{S}^\top \operatorname{diag}\Big(\operatorname{softmax}\big(\boldsymbol{S}_{(j,:)}\boldsymbol{W}_{(:,i)}^Q\boldsymbol{W}_{(:,i)}^{K\top}\boldsymbol{S}^\top/\sqrt{d}\big)\Big)\boldsymbol{S}\boldsymbol{W}^V \\[2mm]
&\quad -\frac{1}{\sqrt{d}}\boldsymbol{W}_{(:,i)}^{K\top}\boldsymbol{S}^\top\Big(\operatorname{softmax}\big(\boldsymbol{S}_{(j,:)}\boldsymbol{W}_{(:,i)}^Q\boldsymbol{W}_{(:,i)}^{K\top}\boldsymbol{S}^\top/\sqrt{d}\big)\Big)^\top\operatorname{softmax}\big(\boldsymbol{S}_{(j,:)}\boldsymbol{W}_{(:,i)}^Q\boldsymbol{W}_{(:,i)}^{K\top}\boldsymbol{S}^\top/\sqrt{d}\big)\boldsymbol{S}\boldsymbol{W}^V\Bigg).
\end{aligned} \tag{27}
$$

The right-hand side of Equation 27 can be rewritten as

$$
\underbrace{\boldsymbol{S}_{(j,:)}}_{\text{size: } 1\times d} \cdot \left( \underbrace{d^{-1/2}}_{1\times 1} \underbrace{\boldsymbol{W}_{(:,i)}^{K}{}^{\top}}_{1\times d} \overbrace{\boldsymbol{S}^{\top}}^{d\times S} \mathrm{diag}\Big( \mathrm{softmax}\big( \overbrace{\boldsymbol{S}_{(j,:)}}^{1\times d} \overbrace{\boldsymbol{W}_{(:,i)}^{Q}}^{d\times 1} \overbrace{\boldsymbol{W}_{(:,i)}^{K}{}^{\top}}^{1\times d} \overbrace{\boldsymbol{S}^{\top}}^{d\times S} /\sqrt{d} \big) \Big) }_{S\times S} \underbrace{\boldsymbol{S}}_{S\times d} \underbrace{\boldsymbol{W}^{V}}_{d\times 1}
$$
$$
-\underbrace{d^{-1/2}}_{1\times 1} \underbrace{\boldsymbol{W}_{(:,i)}^{K}{}^{\top}}_{1\times d} \overbrace{\boldsymbol{S}^{\top}}^{d\times S} \underbrace{\Big( \mathrm{softmax}\big( \overbrace{\boldsymbol{S}_{(j,:)}}^{1\times d} \overbrace{\boldsymbol{W}_{(:,i)}^{Q}}^{d\times 1} \overbrace{\boldsymbol{W}_{(:,i)}^{K}{}^{\top}}^{1\times d} \overbrace{\boldsymbol{S}^{\top}}^{d\times S} /\sqrt{d} \big) \Big)^{\top}}_{S\times 1} \underbrace{\mathrm{softmax}\big( \overbrace{\boldsymbol{S}_{(j,:)}}^{1\times d} \overbrace{\boldsymbol{W}_{(:,i)}^{Q}}^{d\times 1} \overbrace{\boldsymbol{W}_{(:,i)}^{K}{}^{\top}}^{1\times d} \overbrace{\boldsymbol{S}^{\top}}^{d\times S} /\sqrt{d} \big)}_{1\times S} \underbrace{\boldsymbol{S}}_{S\times d} \underbrace{\boldsymbol{W}^{V}}_{d\times 1} \Bigg)
$$

$$
= \underbrace{\boldsymbol{S}_{(j,:)}}_{1\times d} \cdot \left( \underbrace{d^{-1/2}}_{1\times 1} \underbrace{\mathcal{K}_{(:,i)}{}^{\top}}_{1\times S} \mathrm{diag}\Big( \mathrm{softmax}\big( \overbrace{\mathcal{Q}_{(j,i)}}^{1\times 1} \overbrace{\mathcal{K}_{(:,i)}{}^{\top}}^{1\times S} /\sqrt{d} \big) \Big) }_{S\times S} \underbrace{\mathcal{V}}_{S\times 1}
$$
$$
- \underbrace{d^{-1/2}}_{1\times 1} \underbrace{\mathcal{K}_{(:,i)}{}^{\top}}_{1\times S} \underbrace{\Big( \mathrm{softmax}\big( \overbrace{\mathcal{Q}_{(j,i)}}^{1\times 1} \overbrace{\mathcal{K}_{(:,i)}{}^{\top}}^{1\times S} /\sqrt{d} \big) \Big)^{\top}}_{S\times 1} \underbrace{\mathrm{softmax}\big( \overbrace{\mathcal{Q}_{(j,i)}}^{1\times 1} \overbrace{\mathcal{K}_{(:,i)}{}^{\top}}^{1\times S} /\sqrt{d} \big)}_{1\times S} \underbrace{\mathcal{V}}_{S\times 1} \Bigg)
$$

$$
= d^{-1/2} \underbrace{\boldsymbol{S}_{(j,:)}}_{1\times d} \cdot \underbrace{\left( \overbrace{\mathcal{K}_{(:,i)}{}^{\top}}^{1\times S} \overbrace{\mathrm{diag}\big( \mathrm{softmax}(\mathcal{Q}_{(j,i)}\mathcal{K}_{(:,i)}{}^{\top}/\sqrt{d}) \big)}^{S\times S} \overbrace{\mathcal{V}}^{S\times 1} - \overbrace{\mathcal{K}_{(:,i)}{}^{\top}}^{1\times S} \overbrace{\big( \mathrm{softmax}(\mathcal{Q}_{(j,i)}\mathcal{K}_{(:,i)}{}^{\top}/\sqrt{d}) \big)^{\top} \mathrm{softmax}(\mathcal{Q}_{(j,i)}\mathcal{K}_{(:,i)}{}^{\top}/\sqrt{d})}^{S\times S} \overbrace{\mathcal{V}}^{S\times 1} \right)}_{1\times 1} .
$$

$$(28)$$

By combining Equation 26, 27 and 28, we derive

$$
\frac{\partial f_{\theta}(\boldsymbol{S})}{\partial \boldsymbol{W}_{(:,i)}^{Q}}
$$

$$
= \begin{bmatrix} d^{-1/2} \underbrace{\boldsymbol{S}_{(1,:)}}_{1\times d} \cdot \underbrace{\left( \overbrace{\mathcal{K}_{(:,i)}{}^{\top}}^{1\times S} \overbrace{\mathrm{diag}\big( \mathrm{softmax}(\mathcal{Q}_{(1,i)}\mathcal{K}_{(:,i)}{}^{\top}/\sqrt{d}) \big)}^{S\times S} \overbrace{\mathcal{V}}^{S\times 1} - \overbrace{\mathcal{K}_{(:,i)}{}^{\top}}^{1\times S} \overbrace{\big( \mathrm{softmax}(\mathcal{Q}_{(1,i)}\mathcal{K}_{(:,i)}{}^{\top}/\sqrt{d}) \big)^{\top} \mathrm{softmax}(\mathcal{Q}_{(1,i)}\mathcal{K}_{(:,i)}{}^{\top}/\sqrt{d})}^{S\times S} \overbrace{\mathcal{V}}^{S\times 1} \right)}_{1\times 1} \\ \cdots \\ d^{-1/2} \underbrace{\boldsymbol{S}_{(S,:)}}_{1\times d} \cdot \underbrace{\left( \overbrace{\mathcal{K}_{(:,i)}{}^{\top}}^{1\times S} \overbrace{\mathrm{diag}\big( \mathrm{softmax}(\mathcal{Q}_{(S,i)}\mathcal{K}_{(:,i)}{}^{\top}/\sqrt{d}) \big)}^{S\times S} \overbrace{\mathcal{V}}^{S\times 1} - \overbrace{\mathcal{K}_{(:,i)}{}^{\top}}^{1\times S} \overbrace{\big( \mathrm{softmax}(\mathcal{Q}_{(S,i)}\mathcal{K}_{(:,i)}{}^{\top}/\sqrt{d}) \big)^{\top} \mathrm{softmax}(\mathcal{Q}_{(S,i)}\mathcal{K}_{(:,i)}{}^{\top}/\sqrt{d})}^{S\times S} \overbrace{\mathcal{V}}^{S\times 1} \right)}_{1\times 1} \end{bmatrix}_{S\times d}
$$

$$
= \left[ d^{-1/2} \boldsymbol{S}_{(j,:)} \cdot \left( \mathcal{K}_{(:,i)}{}^{\top} \mathrm{diag}\big( \mathrm{softmax}(\mathcal{Q}_{(j,i)}\mathcal{K}_{(:,i)}{}^{\top}/\sqrt{d}) \big) \mathcal{V} - \mathcal{K}_{(:,i)}{}^{\top} \big( \mathrm{softmax}(\mathcal{Q}_{(j,i)}\mathcal{K}_{(:,i)}{}^{\top}/\sqrt{d}) \big)^{\top} \mathrm{softmax}(\mathcal{Q}_{(j,i)}\mathcal{K}_{(:,i)}{}^{\top}/\sqrt{d})\mathcal{V} \right) \right]_{S\times d}
$$

$$
= \mathrm{diag}\Bigg( \mathcal{K}_{(:,i)}{}^{\top} \mathrm{diag}\big( \mathrm{softmax}(\mathcal{Q}_{(1,i)}\mathcal{K}_{(:,i)}{}^{\top}/\sqrt{d}) \big)\mathcal{V} - \mathcal{K}_{(:,i)}{}^{\top} \big( \mathrm{softmax}(\mathcal{Q}_{(1,i)}\mathcal{K}_{(:,i)}{}^{\top}/\sqrt{d}) \big)^{\top} \mathrm{softmax}(\mathcal{Q}_{(1,i)}\mathcal{K}_{(:,i)}{}^{\top}/\sqrt{d})\mathcal{V},
$$
$$
\ldots , \mathcal{K}_{(:,i)}{}^{\top} \mathrm{diag}\big( \mathrm{softmax}(\mathcal{Q}_{(S,i)}\mathcal{K}_{(:,i)}{}^{\top}/\sqrt{d}) \big)\mathcal{V} - \mathcal{K}_{(:,i)}{}^{\top} \big( \mathrm{softmax}(\mathcal{Q}_{(S,i)}\mathcal{K}_{(:,i)}{}^{\top}/\sqrt{d}) \big)^{\top} \mathrm{softmax}(\mathcal{Q}_{(S,i)}\mathcal{K}_{(:,i)}{}^{\top}/\sqrt{d})\mathcal{V} \Bigg) \boldsymbol{S}/\sqrt{d} .
$$

$$(29)$$

The derivative of $f_\theta(\boldsymbol{S})$ with respect to the key weight matrix is derived in a manner similar to that of the query weight matrix. For $i \in \mathbb{N}_p$,

$$
\begin{aligned}
&\frac{\partial f_\theta(\boldsymbol{S})}{\partial \boldsymbol{W}_{(:,i)}^K} \\
&= \frac{\partial \operatorname{softmax}\left(\frac{\mathcal{Q}(\boldsymbol{S})\mathcal{K}(\boldsymbol{S})^\top}{\sqrt{d}}\right)\mathcal{V}(\boldsymbol{S})}{\partial \boldsymbol{W}_{(:,i)}^K} \\
&= \frac{\partial \operatorname{softmax}\left(\frac{\boldsymbol{S}\boldsymbol{W}^Q {\boldsymbol{W}^K}^\top \boldsymbol{S}^\top}{\sqrt{d}}\right)\boldsymbol{S}\boldsymbol{W}^V}{\partial \boldsymbol{W}_{(:,i)}^K} \\
&= \frac{\partial \operatorname{softmax}\left(\frac{\boldsymbol{S}\boldsymbol{W}^Q (\boldsymbol{W}^K \boldsymbol{e}_i \boldsymbol{e}_i^\top)^\top \boldsymbol{S}^\top}{\sqrt{d}}\right)\boldsymbol{S}\boldsymbol{W}^V}{\partial \boldsymbol{W}_{(:,i)}^K} \\
&= \begin{bmatrix} \dfrac{\partial \operatorname{softmax}\left(\frac{\boldsymbol{S}_{(1,:)}\boldsymbol{W}_{(:,i)}^Q {\boldsymbol{W}_{(:,i)}^K}^\top \boldsymbol{S}^\top}{\sqrt{d}}\right)\boldsymbol{S}\boldsymbol{W}^V}{\partial \boldsymbol{W}_{(:,i)}^K} \\ \cdots \\ \dfrac{\partial \operatorname{softmax}\left(\frac{\boldsymbol{S}_{(S,:)}\boldsymbol{W}_{(:,i)}^Q {\boldsymbol{W}_{(:,i)}^K}^\top \boldsymbol{S}^\top}{\sqrt{d}}\right)\boldsymbol{S}\boldsymbol{W}^V}{\partial \boldsymbol{W}_{(:,i)}^K} \end{bmatrix} \\
&= \begin{bmatrix} \partial\dfrac{\boldsymbol{S}_{(1,:)}\boldsymbol{W}_{(:,i)}^Q {\boldsymbol{W}_{(:,i)}^K}^\top \boldsymbol{S}^\top}{\sqrt{d}}\Big/\partial \boldsymbol{W}_{(:,i)}^K \cdot \dfrac{\partial \operatorname{softmax}\left(\frac{\boldsymbol{S}_{(1,:)}\boldsymbol{W}_{(:,i)}^Q {\boldsymbol{W}_{(:,i)}^K}^\top \boldsymbol{S}^\top}{\sqrt{d}}\right)}{\partial\frac{\boldsymbol{S}_{(1,:)}\boldsymbol{W}_{(:,i)}^Q {\boldsymbol{W}_{(:,i)}^K}^\top \boldsymbol{S}^\top}{\sqrt{d}}}\boldsymbol{S}\boldsymbol{W}^V \\ \cdots \\ \partial\dfrac{\boldsymbol{S}_{(S,:)}\boldsymbol{W}_{(:,i)}^Q {\boldsymbol{W}_{(:,i)}^K}^\top \boldsymbol{S}^\top}{\sqrt{d}}\Big/\partial \boldsymbol{W}_{(:,i)}^K \cdot \dfrac{\partial \operatorname{softmax}\left(\frac{\boldsymbol{S}_{(S,:)}\boldsymbol{W}_{(:,i)}^Q {\boldsymbol{W}_{(:,i)}^K}^\top \boldsymbol{S}^\top}{\sqrt{d}}\right)}{\partial\frac{\boldsymbol{S}_{(S,:)}\boldsymbol{W}_{(:,i)}^Q {\boldsymbol{W}_{(:,i)}^K}^\top \boldsymbol{S}^\top}{\sqrt{d}}}\boldsymbol{S}\boldsymbol{W}^V \end{bmatrix} .
\end{aligned}
\tag{30}
$$

Similarly, let's examine it row by row. For the $j$-th row ($j \in \mathbb{N}_S$) of Equation 30, it is

$$
\begin{aligned}
&\frac{\partial\frac{\boldsymbol{S}_{(j,:)}\boldsymbol{W}_{(:,i)}^Q {\boldsymbol{W}_{(:,i)}^K}^\top \boldsymbol{S}^\top}{\sqrt{d}}}{\partial \boldsymbol{W}_{(:,i)}^K}\frac{\partial \operatorname{softmax}\left(\frac{\boldsymbol{S}_{(j,:)}\boldsymbol{W}_{(:,i)}^Q {\boldsymbol{W}_{(:,i)}^K}^\top \boldsymbol{S}^\top}{\sqrt{d}}\right)}{\partial\frac{\boldsymbol{S}_{(j,:)}\boldsymbol{W}_{(:,i)}^Q {\boldsymbol{W}_{(:,i)}^K}^\top \boldsymbol{S}^\top}{\sqrt{d}}}\boldsymbol{S}\boldsymbol{W}^V \\
&= \boldsymbol{S}_{(j,:)}\cdot \frac{1}{\sqrt{d}}{\boldsymbol{W}_{(:,i)}^Q}^\top \boldsymbol{S}^\top \frac{\partial \operatorname{softmax}\left(\frac{\boldsymbol{S}_{(j,:)}\boldsymbol{W}_{(:,i)}^Q {\boldsymbol{W}_{(:,i)}^K}^\top \boldsymbol{S}^\top}{\sqrt{d}}\right)}{\partial\frac{\boldsymbol{S}_{(j,:)}\boldsymbol{W}_{(:,i)}^Q {\boldsymbol{W}_{(:,i)}^K}^\top \boldsymbol{S}^\top}{\sqrt{d}}}\boldsymbol{S}\boldsymbol{W}^V \\
&= \boldsymbol{S}_{(j,:)}\cdot \frac{1}{\sqrt{d}}{\boldsymbol{W}_{(:,i)}^Q}^\top \boldsymbol{S}^\top \frac{\partial\left(\frac{\exp\left(d^{-1/2}\boldsymbol{S}_{(j,:)}\boldsymbol{W}_{(:,i)}^Q {\boldsymbol{W}_{(:,i)}^K}^\top \boldsymbol{S}^\top\right)}{\mathbf{1}^\top \exp\left(d^{-1/2}\boldsymbol{S}_{(j,:)}\boldsymbol{W}_{(:,i)}^Q {\boldsymbol{W}_{(:,i)}^K}^\top \boldsymbol{S}^\top\right)}\right)}{\partial d^{-1/2}\boldsymbol{S}_{(j,:)}\boldsymbol{W}_{(:,i)}^Q {\boldsymbol{W}_{(:,i)}^K}^\top \boldsymbol{S}^\top}\boldsymbol{S}\boldsymbol{W}^V \\
&= \boldsymbol{S}_{(j,:)}\cdot \left(\frac{1}{\sqrt{d}}{\boldsymbol{W}_{(:,i)}^Q}^\top \boldsymbol{S}^\top \operatorname{diag}\left(\operatorname{softmax}\left(\boldsymbol{S}_{(j,:)}\boldsymbol{W}_{(:,i)}^Q {\boldsymbol{W}_{(:,i)}^K}^\top \boldsymbol{S}^\top/\sqrt{d}\right)\right)\boldsymbol{S}\boldsymbol{W}^V \right. \\
&\quad \left. -\frac{1}{\sqrt{d}}{\boldsymbol{W}_{(:,i)}^Q}^\top \boldsymbol{S}^\top \left(\operatorname{softmax}\left(\boldsymbol{S}_{(j,:)}\boldsymbol{W}_{(:,i)}^Q {\boldsymbol{W}_{(:,i)}^K}^\top \boldsymbol{S}^\top/\sqrt{d}\right)\right)^\top \operatorname{softmax}\left(\boldsymbol{S}_{(j,:)}\boldsymbol{W}_{(:,i)}^Q {\boldsymbol{W}_{(:,i)}^K}^\top \boldsymbol{S}^\top/\sqrt{d}\right)\boldsymbol{S}\boldsymbol{W}^V\right).
\end{aligned}
\tag{31}
$$

The right-hand side of Equation 31 can be simplified to:

$$
\underbrace{\boldsymbol{S}_{(j,:)}}_{\text{size: }1\times d} \cdot \Bigg( \underbrace{d^{-1/2}}_{1\times 1} \underbrace{\boldsymbol{W}^{Q}_{(:,i)}{}^{\top}}_{1\times d} \overbrace{\boldsymbol{S}^{\top}}^{d\times S} \mathrm{diag}\Big( \mathrm{softmax}\big( \overbrace{\boldsymbol{S}_{(j,:)}}^{1\times d} \overbrace{\boldsymbol{W}^{Q}_{(:,i)}}^{d\times 1} \overbrace{\boldsymbol{W}^{K}_{(:,i)}{}^{\top}}^{1\times d} \overbrace{\boldsymbol{S}^{\top}}^{d\times S} /\sqrt{d}\big)\Big) \underbrace{\underbrace{\boldsymbol{S}}_{S\times d} \underbrace{\boldsymbol{W}^{V}}_{d\times 1}}_{S\times S}
$$

$$
-\underbrace{d^{-1/2}}_{1\times 1}\underbrace{\boldsymbol{W}^{Q}_{(:,i)}{}^{\top}}_{1\times d}\overbrace{\boldsymbol{S}^{\top}}^{d\times S}\underbrace{\Big(\mathrm{softmax}\big(\overbrace{\boldsymbol{S}_{(j,:)}}^{1\times d}\overbrace{\boldsymbol{W}^{Q}_{(:,i)}}^{d\times 1}\overbrace{\boldsymbol{W}^{K}_{(:,i)}{}^{\top}}^{1\times d}\overbrace{\boldsymbol{S}^{\top}}^{d\times S}/\sqrt{d}\big)\Big)^{\top}}_{S\times 1}\underbrace{\mathrm{softmax}\big(\overbrace{\boldsymbol{S}_{(j,:)}}^{1\times d}\overbrace{\boldsymbol{W}^{Q}_{(:,i)}}^{d\times 1}\overbrace{\boldsymbol{W}^{K}_{(:,i)}{}^{\top}}^{1\times d}\overbrace{\boldsymbol{S}^{\top}}^{d\times S}/\sqrt{d}\big)}_{1\times S}\underbrace{\underbrace{\boldsymbol{S}}_{S\times d}\underbrace{\boldsymbol{W}^{V}}_{d\times 1}}_{}\Bigg)
$$

$$
= \underbrace{\boldsymbol{S}_{(j,:)}}_{1\times d} \cdot \Bigg( \underbrace{d^{-1/2}}_{1\times 1} \underbrace{\mathcal{Q}_{(:,i)}{}^{\top}}_{1\times S} \underbrace{\mathrm{diag}\Big(\mathrm{softmax}\big(\overbrace{\mathcal{Q}_{(j,i)}}^{1\times 1}\overbrace{\mathcal{K}_{(:,i)}{}^{\top}}^{1\times S}/\sqrt{d}\big)\Big)}_{S\times S} \underbrace{\mathcal{V}}_{S\times 1}
$$

$$
-\underbrace{d^{-1/2}}_{1\times 1}\underbrace{\mathcal{Q}_{(:,i)}{}^{\top}}_{1\times S}\underbrace{\Big(\mathrm{softmax}\big(\overbrace{\mathcal{Q}_{(j,i)}}^{1\times 1}\overbrace{\mathcal{K}_{(:,i)}{}^{\top}}^{1\times S}/\sqrt{d}\big)\Big)^{\top}}_{S\times 1}\underbrace{\mathrm{softmax}\big(\overbrace{\mathcal{Q}_{(j,i)}}^{1\times 1}\overbrace{\mathcal{K}_{(:,i)}{}^{\top}}^{1\times S}/\sqrt{d}\big)}_{1\times S}\underbrace{\mathcal{V}}_{S\times 1}\Bigg)
$$

$$
= d^{-1/2}\underbrace{\boldsymbol{S}_{(j,:)}}_{1\times d}\cdot\underbrace{\Bigg(\underbrace{\mathcal{Q}_{(:,i)}{}^{\top}}_{1\times S}\overbrace{\mathrm{diag}\big(\mathrm{softmax}(\mathcal{Q}_{(j,i)}\mathcal{K}_{(:,i)}{}^{\top}/\sqrt{d})\big)}^{S\times S}\overbrace{\mathcal{V}}^{S\times 1}-\underbrace{\mathcal{Q}_{(:,i)}{}^{\top}}_{1\times S}\overbrace{\big(\mathrm{softmax}(\mathcal{Q}_{(j,i)}\mathcal{K}_{(:,i)}{}^{\top}/\sqrt{d})\big)^{\top}\mathrm{softmax}(\mathcal{Q}_{(j,i)}\mathcal{K}_{(:,i)}{}^{\top}/\sqrt{d})}^{S\times S}\overbrace{\mathcal{V}}^{S\times 1}\Bigg)}_{1\times 1} .
$$

$$\tag{32}$$

By merging Equation 30, 31 and 32, we get:

$$
\frac{\partial f_{\theta}(\boldsymbol{S})}{\partial \boldsymbol{W}^{K}_{(:,i)}}
$$

$$
= \begin{bmatrix} d^{-1/2}\underbrace{\boldsymbol{S}_{(1,:)}}_{1\times d}\cdot\underbrace{\Big(\underbrace{\mathcal{Q}_{(:,i)}{}^{\top}}_{1\times S}\overbrace{\mathrm{diag}\big(\mathrm{softmax}(\mathcal{Q}_{(1,i)}\mathcal{K}_{(:,i)}{}^{\top}/\sqrt{d})\big)}^{S\times S}\overbrace{\mathcal{V}}^{S\times 1}-\underbrace{\mathcal{Q}_{(:,i)}{}^{\top}}_{1\times S}\overbrace{\big(\mathrm{softmax}(\mathcal{Q}_{(1,i)}\mathcal{K}_{(:,i)}{}^{\top}/\sqrt{d})\big)^{\top}\mathrm{softmax}(\mathcal{Q}_{(1,i)}\mathcal{K}_{(:,i)}{}^{\top}/\sqrt{d})}^{S\times S}\overbrace{\mathcal{V}}^{S\times 1}\Big)}_{1\times 1} \\ \dots \\ d^{-1/2}\underbrace{\boldsymbol{S}_{(S,:)}}_{1\times d}\cdot\underbrace{\Big(\underbrace{\mathcal{Q}_{(:,i)}{}^{\top}}_{1\times S}\overbrace{\mathrm{diag}\big(\mathrm{softmax}(\mathcal{Q}_{(S,i)}\mathcal{K}_{(:,i)}{}^{\top}/\sqrt{d})\big)}^{S\times S}\overbrace{\mathcal{V}}^{S\times 1}-\underbrace{\mathcal{Q}_{(:,i)}{}^{\top}}_{1\times S}\overbrace{\big(\mathrm{softmax}(\mathcal{Q}_{(S,i)}\mathcal{K}_{(:,i)}{}^{\top}/\sqrt{d})\big)^{\top}\mathrm{softmax}(\mathcal{Q}_{(S,i)}\mathcal{K}_{(:,i)}{}^{\top}/\sqrt{d})}^{S\times S}\overbrace{\mathcal{V}}^{S\times 1}\Big)}_{1\times 1} \end{bmatrix}_{S\times d}
$$

$$
= \Big[ d^{-1/2}\boldsymbol{S}_{(j,:)}\cdot\Big(\mathcal{Q}_{(:,i)}{}^{\top}\mathrm{diag}\big(\mathrm{softmax}(\mathcal{Q}_{(j,i)}\mathcal{K}_{(:,i)}{}^{\top}/\sqrt{d})\big)\mathcal{V}-\mathcal{Q}_{(:,i)}{}^{\top}\big(\mathrm{softmax}(\mathcal{Q}_{(j,i)}\mathcal{K}_{(:,i)}{}^{\top}/\sqrt{d})\big)^{\top}\mathrm{softmax}(\mathcal{Q}_{(j,i)}\mathcal{K}_{(:,i)}{}^{\top}/\sqrt{d})\mathcal{V}\Big)\Big]_{S\times d}
$$

$$
= \mathrm{diag}\Bigg( \mathcal{Q}_{(:,i)}{}^{\top}\mathrm{diag}\big(\mathrm{softmax}(\mathcal{Q}_{(1,i)}\mathcal{K}_{(:,i)}{}^{\top}/\sqrt{d})\big)\mathcal{V}-\mathcal{Q}_{(:,i)}{}^{\top}\big(\mathrm{softmax}(\mathcal{Q}_{(1,i)}\mathcal{K}_{(:,i)}{}^{\top}/\sqrt{d})\big)^{\top}\mathrm{softmax}(\mathcal{Q}_{(1,i)}\mathcal{K}_{(:,i)}{}^{\top}/\sqrt{d})\mathcal{V},
$$

$$
\dots, \mathcal{Q}_{(:,i)}{}^{\top}\mathrm{diag}\big(\mathrm{softmax}(\mathcal{Q}_{(S,i)}\mathcal{K}_{(:,i)}{}^{\top}/\sqrt{d})\big)\mathcal{V}-\mathcal{Q}_{(:,i)}{}^{\top}\big(\mathrm{softmax}(\mathcal{Q}_{(S,i)}\mathcal{K}_{(:,i)}{}^{\top}/\sqrt{d})\big)^{\top}\mathrm{softmax}(\mathcal{Q}_{(S,i)}\mathcal{K}_{(:,i)}{}^{\top}/\sqrt{d})\mathcal{V} \Bigg)\boldsymbol{S}/\sqrt{d} .
$$

$$\tag{33}$$

From Equations 25, 29, and 33, it can be observed that the ANN gradient for a single sequence resembles that for a batch of feature vector inputs, as the gradient can be decomposed for each element. This demonstrates the parallelization-friendly nature of the attention mechanism from a gradient perspective.

These results can be directly extended to the case where each component of the output $f_{\theta^t}(\boldsymbol{S})_{(j,:)}$ is a vector, by considering a multi-dimensional setting (Zhang et al., 2023a). The extension to multi-head cases can be done by broadcasting, which involves repeating the derivation in parallel as many times as there are heads.

## A.4 ATTENTION NEURAL TANGENT KERNEL (ANTK)

By incorporating the parameter evolution (*i.e.*, Equation 7)

$$
\frac{\partial \theta^t}{\partial t} = -\frac{\eta}{NS}\left[\frac{\partial\mathcal{L}(f_{\theta^t}(\boldsymbol{S}_1),\boldsymbol{y}_1)}{\partial f_{\theta^t}(\boldsymbol{S}_1)},\dots,\frac{\partial\mathcal{L}(f_{\theta^t}(\boldsymbol{S}_N),\boldsymbol{y}_N)}{\partial f_{\theta^t}(\boldsymbol{S}_N)}\right]\cdot\left[\frac{\partial f_{\theta^t}(\boldsymbol{S}_i)}{\partial\theta^t}\right]_N . \tag{34}
$$

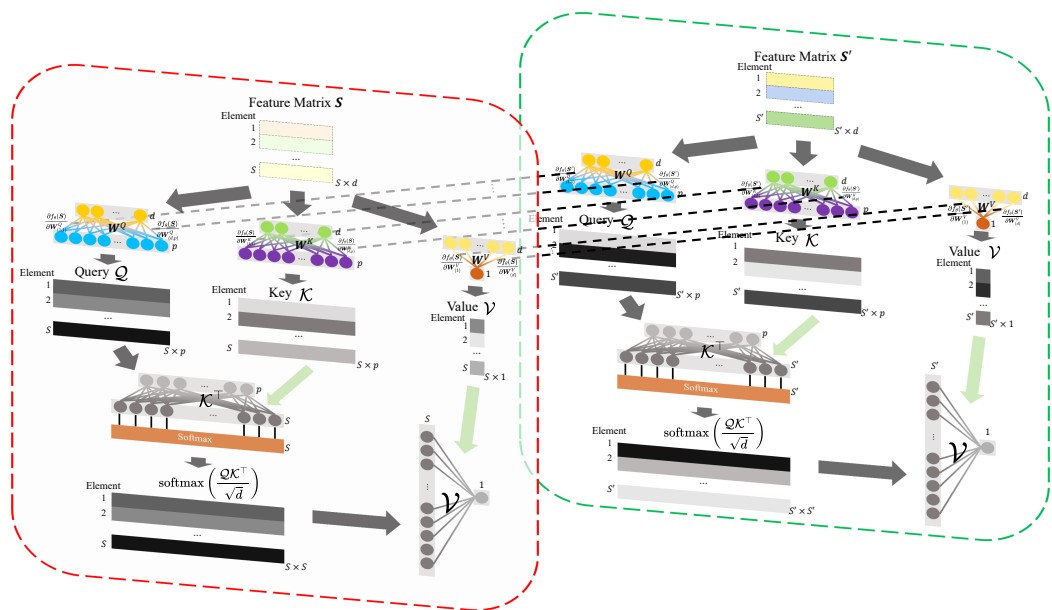

Figure 3: Graphical depiction of the ANTK computation process: $K_\theta(\boldsymbol{S}_S, \boldsymbol{S}'_{S'}) = \left\langle \frac{\partial f_\theta(\boldsymbol{S})}{\partial \theta}, \frac{\partial f_\theta(\boldsymbol{S}')}{\partial \theta} \right\rangle =$

$$\left[ \frac{\partial f_\theta(\boldsymbol{S})_{(i,:)}}{\partial \boldsymbol{W}^V_{(1)}} \frac{\partial f_\theta(\boldsymbol{S}')_{(j,:)}}{\partial \boldsymbol{W}^V_{(1)}} + \cdots + \frac{\partial f_\theta(\boldsymbol{S})_{(i,:)}}{\partial \boldsymbol{W}^V_{(d)}} \frac{\partial f_\theta(\boldsymbol{S}')_{(j,:)}}{\partial \boldsymbol{W}^V_{(d)}} + \frac{\partial f_\theta(\boldsymbol{S})_{(i,:)}}{\partial \boldsymbol{W}^Q_{(1,1)}} \frac{\partial f_\theta(\boldsymbol{S}')_{(j,:)}}{\partial \boldsymbol{W}^Q_{(1,1)}} + \cdots + \frac{\partial f_\theta(\boldsymbol{S})_{(i,:)}}{\partial \boldsymbol{W}^Q_{(d,p)}} \frac{\partial f_\theta(\boldsymbol{S}')_{(j,:)}}{\partial \boldsymbol{W}^Q_{(d,p)}} + \frac{\partial f_\theta(\boldsymbol{S})_{(i,:)}}{\partial \boldsymbol{W}^K_{(1,1)}} \frac{\partial f_\theta(\boldsymbol{S}')_{(j,:)}}{\partial \boldsymbol{W}^K_{(1,1)}} + \cdots + \frac{\partial f_\theta(\boldsymbol{S})_{(i,:)}}{\partial \boldsymbol{W}^K_{(d,p)}} \frac{\partial f_\theta(\boldsymbol{S}')_{(j,:)}}{\partial \boldsymbol{W}^K_{(d,p)}} \right]_{S \times S'; i \in \mathbb{N}_S, j \in \mathbb{N}_{S'}}.$$

into the first-order approximation term $(*)$ of Equation 14, we derive

$$
\begin{aligned}
(*) &= \left\langle \frac{\partial f_{\theta^t}(\cdot)}{\partial \theta^t}, -\frac{\eta}{NS} \left[ \frac{\partial \mathcal{L}(f_{\theta^t}(\boldsymbol{S}_1), \boldsymbol{y}_1)}{\partial f_{\theta^t}(\boldsymbol{S}_1)}, \ldots, \frac{\partial \mathcal{L}(f_{\theta^t}(\boldsymbol{S}_N), \boldsymbol{y}_N)}{\partial f_{\theta^t}(\boldsymbol{S}_N)} \right] \cdot \left[ \frac{\partial f_{\theta^t}(\boldsymbol{S}_i)}{\partial \theta^t} \right]_N \right\rangle \\
&= -\frac{\eta}{NS} \left[ \frac{\partial \mathcal{L}(f_{\theta^t}(\boldsymbol{S}_1), \boldsymbol{y}_1)}{\partial f_{\theta^t}(\boldsymbol{S}_1)}, \ldots, \frac{\partial \mathcal{L}(f_{\theta^t}(\boldsymbol{S}_N), \boldsymbol{y}_N)}{\partial f_{\theta^t}(\boldsymbol{S}_N)} \right] \cdot \left\langle \frac{\partial f_{\theta^t}(\cdot)}{\partial \theta^t}, \left[ \frac{\partial f_{\theta^t}(\boldsymbol{S}_i)}{\partial \theta^t} \right]_N \right\rangle \\
&= -\frac{\eta}{NS} \left[ \frac{\partial \mathcal{L}(f_{\theta^t}(\boldsymbol{S}_1), \boldsymbol{y}_1)}{\partial f_{\theta^t}(\boldsymbol{S}_1)}, \ldots, \frac{\partial \mathcal{L}(f_{\theta^t}(\boldsymbol{S}_N), \boldsymbol{y}_N)}{\partial f_{\theta^t}(\boldsymbol{S}_N)} \right] \cdot \left[ \left\langle \frac{\partial f_{\theta^t}(\cdot)}{\partial \theta^t}, \frac{\partial f_{\theta^t}(\boldsymbol{S}_i)}{\partial \theta^t} \right\rangle \right]_N \\
&= -\frac{\eta}{NS} \left[ \frac{\partial \mathcal{L}(f_{\theta^t}(\boldsymbol{S}_1), \boldsymbol{y}_1)}{\partial f_{\theta^t}(\boldsymbol{S}_1)}, \ldots, \frac{\partial \mathcal{L}(f_{\theta^t}(\boldsymbol{S}_N), \boldsymbol{y}_N)}{\partial f_{\theta^t}(\boldsymbol{S}_N)} \right] \cdot \left[ K_{\theta^t}(\boldsymbol{S}_i, \cdot) \right]_N,
\end{aligned}
\tag{35}
$$

which leads to Equation 15 expressed as

$$
\frac{\partial f_{\theta^t}}{\partial t} = -\frac{\eta}{NS} \left[ \frac{\partial \mathcal{L}(f_{\theta^t}(\boldsymbol{S}_1), \boldsymbol{y}_1)}{\partial f_{\theta^t}(\boldsymbol{S}_1)}, \ldots, \frac{\partial \mathcal{L}(f_{\theta^t}(\boldsymbol{S}_N), \boldsymbol{y}_N)}{\partial f_{\theta^t}(\boldsymbol{S}_N)} \right] \cdot \left[ K_{\theta^t}(\boldsymbol{S}_i, \cdot) \right]_N + o\left( \frac{\partial \theta^t}{\partial t} \right),
\tag{36}
$$

where the symmetric and positive definite $K_{\theta^t}(\boldsymbol{S}_i, \cdot) := \left\langle \frac{\partial f_{\theta^t}(\boldsymbol{S}_i)}{\partial \theta^t}, \frac{\partial f_{\theta^t}(\cdot)}{\partial \theta^t} \right\rangle$ is called the attention neural tangent kernel (ANTK) (Jacot et al., 2018; Yang, 2019; Hron et al., 2020). Specifically, ANTK for $\boldsymbol{S}_{(i,:)}$ and $\boldsymbol{S}'_{(j,:)}$ is a scalar $K(\boldsymbol{S}_{(i,:)}, \boldsymbol{S}'_{(j,:)}) = \left\langle \frac{\partial f_{\theta^t}(\boldsymbol{S})_{(i,:)}}{\partial \theta^t}, \frac{\partial f_{\theta^t}(\boldsymbol{S}')_{(j,:)}}{\partial \theta^t} \right\rangle$. Figure 3 illustrates the ANTK computation process, where typically, the length of all training sequences is standardized to the maximum length. In simple terms, examining a model's behavior by focusing on the model itself, rather than its parameters, often involves the use of kernel functions.

The quantity $\frac{\partial f_{\theta^t}(\cdot)}{\partial \theta^t}$, which represents the partial derivative of the ANN with respect to its parameters and appears in $K_{\theta^t}(\boldsymbol{S}_i, \cdot) = \left\langle \frac{\partial f_{\theta^t}(\boldsymbol{S}_i)}{\partial \theta^t}, \frac{\partial f_{\theta^t}(\cdot)}{\partial \theta^t} \right\rangle$, is determined by both the network architecture and the specific parameters $\theta^t$, but it is independent of the input sequences. In contrast, the term $\frac{\partial f_{\theta^t}(\boldsymbol{S}_i)}{\partial \theta^t}$ depends not only on the ANN structure and specific $\theta^t$, but also on the input sequence $\boldsymbol{S}$. When the input for $\frac{\partial f_{\theta^t}(\boldsymbol{S}_i)}{\partial \theta^t}$ is unspecified, the ANTK simplifies to a general form $K_{\theta^t}(\cdot, \cdot)$.

However, when a specific sequence $S_j$ is provided as the input to $\frac{\partial f_{\theta^t}(\cdot)}{\partial \theta^t}$, the ANTK becomes a matrix defined as $K_{\theta^t}(S_i, S_j) = \left\langle \frac{\partial f_{\theta^t}(S_i)}{\partial \theta^t}, \frac{\partial f_{\theta^t}(S_j)}{\partial \theta^t} \right\rangle$. This formulation aligns with the vector-valued kernel used in functional gradient descent (Zhang et al., 2023a). When the input sequence $S_i$ is specified, one argument of $K_{\theta^t}$ is fixed, leading the ANN to update along $K_{\theta^t}(S_i, \cdot)$, with the magnitude of the update determined by $\frac{\partial f_{\theta^t}(S_i)}{\partial \theta^t}$. This process reflects the core mechanism of functional gradient descent. In summary, the ANTK and the canonical vector-valued kernel share a consistent mathematical framework and exhibit similar effects on the evolution of the associated ANN. Additionally, Theorem 3 establishes the asymptotic relationship between the ANTK and the canonical kernel used in functional gradient descent.

## A.5 ATTENT ALGORITHM

---

**Algorithm 1** AtteNT Algorithm

---

**Input:** Target mapping $f^*$ realized by a dense set of sequence-property pairs, initial ANN $f_{\theta^0}$, the size of selected training set $m \leq N$, small constant $\epsilon > 0$ and maximal iteration number $T$

Set $f_{\theta^t} \leftarrow f_{\theta^0}$, $t = 0$

**while** $t \leq T$ and $\|[f_{\theta^t}(S_i) - f^*(S_i)]_N\|_{\mathcal{F}} \geq \epsilon$ **do**

    **The teacher** selects $m$ teaching sequences:

    `/* Sequences associated with the `$m$` largest `$\|f_{\theta^t}(S_i) - f^*(S_i)\|_2$`                    */`

    $\{S_i\}_m^* = \underset{\{S_i\}_m \subseteq \{S_i\}_N}{\arg\max} \|[f_{\theta^t}(S_i) - f^*(S_i)]_m\|_{\mathcal{F}}$

    Provide $\{S_i\}_m^*$ to the attention learner

    **The learner** updates $f_{\theta^t}$ based on received $\{S_i\}_m^*$:

    `// Parameter-based gradient descent`

    $\theta^t \leftarrow \theta^t - \frac{\eta}{mS} \sum_{S_i \in \{S_i\}_m^*} \sum_{j=1}^{S} \nabla_\theta \mathcal{L}(f_{\theta^t}(S_i)_{(j,:)}, f^*(S_i)_{(j,:)})$

    Set $t \leftarrow t + 1$

**end**

---

## B    DETAILED PROOFS

### B.1    PROOF OF THEOREM 3

By examining the evolution of an ANN through changes in its parameters and from a high-level perspective within the function space, we obtain.

$$
\begin{aligned}
&-\frac{\eta}{NS}\left[\frac{\partial\mathcal{L}(f_{\theta^t}(\boldsymbol{S}_1),\boldsymbol{y}_1)}{\partial f_{\theta^t}(\boldsymbol{S}_1)},\ldots,\frac{\partial\mathcal{L}(f_{\theta^t}(\boldsymbol{S}_N),\boldsymbol{y}_N)}{\partial f_{\theta^t}(\boldsymbol{S}_N)}\right][K(\boldsymbol{S}_i,\cdot)]_N\\
=\quad&-\frac{\eta}{NS}\left[\frac{\partial\mathcal{L}(f_{\theta^t}(\boldsymbol{S}_1),\boldsymbol{y}_1)}{\partial f_{\theta^t}(\boldsymbol{S}_1)},\ldots,\frac{\partial\mathcal{L}(f_{\theta^t}(\boldsymbol{S}_N),\boldsymbol{y}_N)}{\partial f_{\theta^t}(\boldsymbol{S}_N)}\right]\cdot\left[\left\langle\frac{\partial f_{\theta^t}(\boldsymbol{S}_i)}{\partial\theta^t},\frac{\partial f_{\theta^t}(\cdot)}{\partial\theta^t}\right\rangle\right]_N+o\left(\frac{\partial\theta^t}{\partial t}\right).
\end{aligned}
$$

$$(37)$$

Upon reorganizing, we derive

$$
-\frac{\eta}{NS}\left[\frac{\partial\mathcal{L}(f_{\theta^t}(\boldsymbol{S}_1),\boldsymbol{y}_1)}{\partial f_{\theta^t}(\boldsymbol{S}_1)},\ldots,\frac{\partial\mathcal{L}(f_{\theta^t}(\boldsymbol{S}_N),\boldsymbol{y}_N)}{\partial f_{\theta^t}(\boldsymbol{S}_N)}\right]\cdot[K(\boldsymbol{S}_i,\cdot)-K_{\theta^t}(\boldsymbol{S}_i,\cdot)]_N=o\left(\frac{\partial\theta^t}{\partial t}\right). \quad (38)
$$

By integrating the parameter evolution

$$
\frac{\partial\theta^t}{\partial t}=-\eta\frac{\partial\mathcal{L}}{\partial\theta^t}=-\frac{\eta}{NS}\left[\frac{\partial\mathcal{L}(f_{\theta^t}(\boldsymbol{S}_1),\boldsymbol{y}_1)}{\partial f_{\theta^t}(\boldsymbol{S}_1)},\ldots,\frac{\partial\mathcal{L}(f_{\theta^t}(\boldsymbol{S}_N),\boldsymbol{y}_N)}{\partial f_{\theta^t}(\boldsymbol{S}_N)}\right]\cdot\left[\frac{\partial f_{\theta^t}(\boldsymbol{S}_i)}{\partial\theta^t}\right]_N \quad (39)
$$

into the remainder, we get

$$
\begin{aligned}
&-\frac{\eta}{NS}\left[\frac{\partial\mathcal{L}(f_{\theta^t}(\boldsymbol{S}_1),\boldsymbol{y}_1)}{\partial f_{\theta^t}(\boldsymbol{S}_1)},\ldots,\frac{\partial\mathcal{L}(f_{\theta^t}(\boldsymbol{S}_N),\boldsymbol{y}_N)}{\partial f_{\theta^t}(\boldsymbol{S}_N)}\right]\cdot[K(\boldsymbol{S}_i,\cdot)-K_{\theta^t}(\boldsymbol{S}_i,\cdot)]_N\\
=\quad&o\left(-\frac{\eta}{NS}\left[\frac{\partial\mathcal{L}(f_{\theta^t}(\boldsymbol{S}_1),\boldsymbol{y}_1)}{\partial f_{\theta^t}(\boldsymbol{S}_1)},\ldots,\frac{\partial\mathcal{L}(f_{\theta^t}(\boldsymbol{S}_N),\boldsymbol{y}_N)}{\partial f_{\theta^t}(\boldsymbol{S}_N)}\right]\cdot\left[\frac{\partial f_{\theta^t}(\boldsymbol{S}_i)}{\partial\theta^t}\right]_N\right).
\end{aligned}
$$

$$(40)$$

When training an ANN with a convex loss $\mathcal{L}$, which is convex in terms of $f_\theta$ but not necessarily in terms of $\theta$, the following limit holds for the vector: $\lim_{t\to\infty}\left[\frac{\partial\mathcal{L}(f_{\theta^t}(\boldsymbol{S}_1),\boldsymbol{y}_1)}{\partial f_{\theta^t}(\boldsymbol{S}_1)},\ldots,\frac{\partial\mathcal{L}(f_{\theta^t}(\boldsymbol{S}_N),\boldsymbol{y}_N)}{\partial f_{\theta^t}(\boldsymbol{S}_N)}\right]=$
$\boldsymbol{0}$. Since the right-hand side of this equation is a higher-order infinitesimal relative to the left, maintaining this equality results in the conclusion that

$$
\lim_{t\to\infty}\left[K(\boldsymbol{S}_i,\cdot)-K_{\theta^t}(\boldsymbol{S}_i,\cdot)\right]_N=\boldsymbol{0}. \quad (41)
$$

This suggests that for each training point, *i.e.*, input sequence $\boldsymbol{S}\in\{\boldsymbol{S}_i\}_N$, ANTK converges pointwise to the canonical kernel.

∎

### B.2    PROOF OF PROPOSITION 4

Referring to the definition of the Fréchet derivative in Definition 2, the convexity of $\mathcal{L}$ implies that

$$
\frac{\partial\mathcal{L}}{\partial t}\leq\underbrace{\left\langle\frac{\partial\mathcal{L}}{\partial f_{\theta^{t+1}}},\frac{f_{\theta^t}}{\partial t}\right\rangle_{\mathcal{H}}}_{\Upsilon}. \quad (42)
$$

By computing the Fréchet derivative of $\frac{\partial \mathcal{L}}{\partial f_{\theta^{t+1}}}$ and the evolution of $f_{\theta^t}$, the term on the right-hand side, $\Upsilon$, can be expressed as

$$
\begin{aligned}
\Upsilon &= \left\langle \mathcal{G}^{t+1}, -\eta \mathcal{G}^t \right\rangle_{\mathcal{H}} \\
&= -\frac{\eta}{N^2 S^2} \left\langle \left[ \frac{\partial \mathcal{L}(f_{\theta^{t+1}}(\boldsymbol{S}_1), \boldsymbol{y}_1)}{\partial f_{\theta^{t+1}}(\boldsymbol{S}_1)}, \ldots, \frac{\partial \mathcal{L}(f_{\theta^{t+1}}(\boldsymbol{S}_N), \boldsymbol{y}_N)}{\partial f_{\theta^{t+1}}(\boldsymbol{S}_N)} \right] \cdot [K_{\boldsymbol{S}_i}]_N \, , \right. \\
&\qquad\qquad \left. [K_{\boldsymbol{S}_i}]_N^\top \cdot \left[ \frac{\partial \mathcal{L}(f_{\theta^t}(\boldsymbol{S}_1), \boldsymbol{y}_1)}{\partial f_{\theta^t}(\boldsymbol{S}_1)}, \ldots, \frac{\partial \mathcal{L}(f_{\theta^t}(\boldsymbol{S}_N), \boldsymbol{y}_N)}{\partial f_{\theta^t}(\boldsymbol{S}_N)} \right]^\top \right\rangle_{\mathcal{H}} \\
&= -\frac{\eta}{N^2 S^2} \left[ \frac{\partial \mathcal{L}(f_{\theta^{t+1}}(\boldsymbol{S}_1), \boldsymbol{y}_1)}{\partial f_{\theta^{t+1}}(\boldsymbol{S}_1)}, \ldots, \frac{\partial \mathcal{L}(f_{\theta^{t+1}}(\boldsymbol{S}_N), \boldsymbol{y}_N)}{\partial f_{\theta^{t+1}}(\boldsymbol{S}_N)} \right] \cdot \left\langle [K_{\boldsymbol{S}_i}]_N, [K_{\boldsymbol{S}_i}]_N^\top \right\rangle_{\mathcal{H}} \\
&\qquad\qquad \cdot \left[ \frac{\partial \mathcal{L}(f_{\theta^t}(\boldsymbol{S}_1), \boldsymbol{y}_1)}{\partial f_{\theta^t}(\boldsymbol{S}_1)}, \ldots, \frac{\partial \mathcal{L}(f_{\theta^t}(\boldsymbol{S}_N), \boldsymbol{y}_N)}{\partial f_{\theta^t}(\boldsymbol{S}_N)} \right]^\top \\
&= -\frac{\eta}{NS} \left[ \frac{\partial \mathcal{L}(f_{\theta^{t+1}}(\boldsymbol{S}_1), \boldsymbol{y}_1)}{\partial f_{\theta^{t+1}}(\boldsymbol{S}_1)}, \ldots, \frac{\partial \mathcal{L}(f_{\theta^{t+1}}(\boldsymbol{S}_N), \boldsymbol{y}_N)}{\partial f_{\theta^{t+1}}(\boldsymbol{S}_N)} \right] \bar{\boldsymbol{K}} \left[ \frac{\partial \mathcal{L}(f_{\theta^t}(\boldsymbol{S}_1), \boldsymbol{y}_1)}{\partial f_{\theta^t}(\boldsymbol{S}_1)}, \ldots, \frac{\partial \mathcal{L}(f_{\theta^t}(\boldsymbol{S}_N), \boldsymbol{y}_N)}{\partial f_{\theta^t}(\boldsymbol{S}_N)} \right]^\top,
\end{aligned}
$$
(43)

where $\bar{\boldsymbol{K}} = \boldsymbol{K}/(NS)$, and $\boldsymbol{K}$ is an $NS \times NS$ symmetric, positive definite block matrix with elements $K(\boldsymbol{S}_i, \boldsymbol{S}_j)$ positioned in the $i$-th row and $j$-th column block. For convenience, we use a simplified column vector notation $\left[ \partial_{f_{\theta\square}} \mathcal{L}(f_{\theta\square}; \boldsymbol{S}_i) \right]_N := \left[ \partial_{f_{\theta\square}} \mathcal{L}(f_{\theta\square}; \boldsymbol{S}_1), \ldots, \partial_{f_{\theta\square}} \mathcal{L}(f_{\theta\square}; \boldsymbol{S}_N) \right]^\top$ with $\partial_{f_{\theta\square}} \mathcal{L}(f_{\theta\square}; \boldsymbol{S}_i) := \frac{\partial \mathcal{L}(f_{\theta\square}(\boldsymbol{S}_i), \boldsymbol{y}_i)}{\partial f_{\theta\square}(\boldsymbol{S}_i)}$. The last term in Equation 43 can then be rewritten as

$$
\begin{aligned}
&-\frac{\eta}{NS} \left[ \partial_{f_{\theta^t}} \mathcal{L}(f_{\theta^t}; \boldsymbol{S}_i) \right]_N^\top \bar{\boldsymbol{K}} \left[ \partial_{f_{\theta^{t+1}}} \mathcal{L}(f_{\theta^{t+1}}; \boldsymbol{S}_i) \right]_N \\
={} &-\frac{\eta}{NS} \left[ \partial_{f_{\theta^t}} \mathcal{L}(f_{\theta^t}; \boldsymbol{S}_i) \right]_N^\top \bar{\boldsymbol{K}} \left( \left[ \partial_{f_{\theta^{t+1}}} \mathcal{L}(f_{\theta^{t+1}}; \boldsymbol{S}_i) \right]_N + \left[ \partial_{f_{\theta^t}} \mathcal{L}(f_{\theta^t}; \boldsymbol{S}_i) \right]_N - \left[ \partial_{f_{\theta^t}} \mathcal{L}(f_{\theta^t}; \boldsymbol{S}_i) \right]_N \right) \\
={} &-\frac{\eta}{NS} \left[ \partial_{f_{\theta^t}} \mathcal{L}(f_{\theta^t}; \boldsymbol{S}_i) \right]_N^\top \bar{\boldsymbol{K}} \left[ \partial_{f_{\theta^t}} \mathcal{L}(f_{\theta^t}; \boldsymbol{S}_i) \right]_N \\
&-\frac{\eta}{NS} \left[ \partial_{f_{\theta^t}} \mathcal{L}(f_{\theta^t}; \boldsymbol{S}_i) \right]_N^\top \bar{\boldsymbol{K}} \left( \left[ \partial_{f_{\theta^{t+1}}} \mathcal{L}(f_{\theta^{t+1}}; \boldsymbol{S}_i) \right]_N - \left[ \partial_{f_{\theta^t}} \mathcal{L}(f_{\theta^t}; \boldsymbol{S}_i) \right]_N \right) \\
={} &-\frac{\eta}{NS} \left[ \partial_{f_{\theta^t}} \mathcal{L}(f_{\theta^t}; \boldsymbol{S}_i) \right]_N^\top \bar{\boldsymbol{K}} \left[ \partial_{f_{\theta^t}} \mathcal{L}(f_{\theta^t}; \boldsymbol{S}_i) \right]_N \\
&+\frac{\eta}{NS} \left( \left[ \partial_{f_{\theta^{t+1}}} \mathcal{L}(f_{\theta^{t+1}}; \boldsymbol{S}_i) \right]_N^\top - \left[ \partial_{f_{\theta^t}} \mathcal{L}(f_{\theta^t}; \boldsymbol{S}_i) \right]_N^\top - \left[ \partial_{f_{\theta^{t+1}}} \mathcal{L}(f_{\theta^{t+1}}; \boldsymbol{S}_i) \right]_N^\top \right) \\
&\cdot \bar{\boldsymbol{K}} \cdot \left( \left[ \partial_{f_{\theta^{t+1}}} \mathcal{L}(f_{\theta^{t+1}}; \boldsymbol{S}_i) \right]_N - \left[ \partial_{f_{\theta^t}} \mathcal{L}(f_{\theta^t}; \boldsymbol{S}_i) \right]_N \right) .
\end{aligned}
$$
(44)

The last term in Equation 44 above can be expanded to

$$
\begin{aligned}
&\frac{\eta}{NS} \left( \left[ \partial_{f_{\theta^{t+1}}} \mathcal{L}(f_{\theta^{t+1}}; \boldsymbol{S}_i) \right]_N^\top - \left[ \partial_{f_{\theta^t}} \mathcal{L}(f_{\theta^t}; \boldsymbol{S}_i) \right]_N^\top - \left[ \partial_{f_{\theta^{t+1}}} \mathcal{L}(f_{\theta^{t+1}}; \boldsymbol{S}_i) \right]_N^\top \right) \\
&\cdot \bar{\boldsymbol{K}} \left( \left[ \partial_{f_{\theta^{t+1}}} \mathcal{L}(f_{\theta^{t+1}}; \boldsymbol{S}_i) \right]_N - \left[ \partial_{f_{\theta^t}} \mathcal{L}(f_{\theta^t}; \boldsymbol{S}_i) \right]_N \right) \\
={} &\frac{\eta}{NS} \left( \left[ \partial_{f_{\theta^{t+1}}} \mathcal{L}(f_{\theta^{t+1}}; \boldsymbol{S}_i) \right]_N - \left[ \partial_{f_{\theta^t}} \mathcal{L}(f_{\theta^t}; \boldsymbol{S}_i) \right]_N \right)^\top \bar{\boldsymbol{K}} \left( \left[ \partial_{f_{\theta^{t+1}}} \mathcal{L}(f_{\theta^{t+1}}; \boldsymbol{S}_i) \right]_N - \left[ \partial_{f_{\theta^t}} \mathcal{L}(f_{\theta^t}; \boldsymbol{S}_i) \right]_N \right) \\
&-\frac{\eta}{NS} \left[ \partial_{f_{\theta^{t+1}}} \mathcal{L}(f_{\theta^{t+1}}; \boldsymbol{S}_i) \right]_N^\top \bar{\boldsymbol{K}} \left( \left[ \partial_{f_{\theta^{t+1}}} \mathcal{L}(f_{\theta^{t+1}}; \boldsymbol{S}_i) \right]_N - \left[ \partial_{f_{\theta^t}} \mathcal{L}(f_{\theta^t}; \boldsymbol{S}_i) \right]_N \right) \\
={} &\frac{\eta}{NS} \left[ \partial_{f_{\theta^{t+1}}} \mathcal{L}(f_{\theta^{t+1}}; \boldsymbol{S}_i) - \partial_{f_{\theta^t}} \mathcal{L}(f_{\theta^t}; \boldsymbol{S}_i) \right]_N^\top \bar{\boldsymbol{K}} \left[ \partial_{f_{\theta^{t+1}}} \mathcal{L}(f_{\theta^{t+1}}; \boldsymbol{S}_i) - \partial_{f_{\theta^t}} \mathcal{L}(f_{\theta^t}; \boldsymbol{S}_i) \right]_N \\
&-\frac{\eta}{NS} \left( \left[ \partial_{f_{\theta^{t+1}}} \mathcal{L}(f_{\theta^{t+1}}; \boldsymbol{S}_i) \right]_N - \frac{1}{2} \left[ \partial_{f_{\theta^t}} \mathcal{L}(f_{\theta^t}; \boldsymbol{S}_i) \right]_N \right)^\top \bar{\boldsymbol{K}} \left( \left[ \partial_{f_{\theta^{t+1}}} \mathcal{L}(f_{\theta^{t+1}}; \boldsymbol{S}_i) \right]_N - \frac{1}{2} \left[ \partial_{f_{\theta^t}} \mathcal{L}(f_{\theta^t}; \boldsymbol{S}_i) \right]_N \right) \\
&+\frac{\eta}{4NS} \left[ \partial_{f_{\theta^t}} \mathcal{L}(f_{\theta^t}; \boldsymbol{S}_i) \right]_N^\top \bar{\boldsymbol{K}} \left[ \partial_{f_{\theta^t}} \mathcal{L}(f_{\theta^t}; \boldsymbol{S}_i) \right]_N .
\end{aligned}
$$
(45)

Given that $\bar{\boldsymbol{K}}$ is positive definite, it follows that

$$
\frac{\eta}{NS} \left( \left[ \partial_{f_{\theta^{t+1}}} \mathcal{L}(f_{\theta^{t+1}}; \boldsymbol{S}_i) \right]_N - \frac{1}{2} \left[ \partial_{f_{\theta^t}} \mathcal{L}(f_{\theta^t}; \boldsymbol{S}_i) \right]_N \right)^\top \bar{\boldsymbol{K}} \left( \left[ \partial_{f_{\theta^{t+1}}} \mathcal{L}(f_{\theta^{t+1}}; \boldsymbol{S}_i) \right]_N - \frac{1}{2} \left[ \partial_{f_{\theta^t}} \mathcal{L}(f_{\theta^t}; \boldsymbol{S}_i) \right]_N \right)
$$

is a non-negative term. Therefore, by merging Equations 43, 44, and 45, we derive

$$
\begin{aligned}
\Upsilon \quad \leq \quad & -\frac{3\eta}{4NS} \underbrace{\left[\partial_{f_{\theta^t}}\mathcal{L}(f_{\theta^t};\boldsymbol{S}_i)\right]_N^\top \bar{\boldsymbol{K}} \left[\partial_{f_{\theta^t}}\mathcal{L}(f_{\theta^t};\boldsymbol{S}_i)\right]_N}_{\Phi} \\
& +\frac{\eta}{NS} \underbrace{\left[\partial_{f_{\theta^{t+1}}}\mathcal{L}(f_{\theta^{t+1}};\boldsymbol{S}_i) - \partial_{f_{\theta^t}}\mathcal{L}(f_{\theta^t};\boldsymbol{S}_i)\right]_N^\top \bar{\boldsymbol{K}} \left[\partial_{f_{\theta^{t+1}}}\mathcal{L}(f_{\theta^{t+1}};\boldsymbol{S}_i) - \partial_{f_{\theta^t}}\mathcal{L}(f_{\theta^t};\boldsymbol{S}_i)\right]_N}_{\Psi} \ .
\end{aligned}
\tag{46}
$$

Based on the definition of the evaluation functional and the assumption that $\mathcal{L}$ is Lipschitz smooth with a constant $\tau > 0$, the term $\Psi$ in the final part of Equation 46 is bounded above as follows:

$$
\begin{aligned}
\Psi \quad = \quad & \left[\partial_{f_{\theta^{t+1}}}\mathcal{L}(f_{\theta^{t+1}};\boldsymbol{S}_i) - \partial_{f_{\theta^t}}\mathcal{L}(f_{\theta^t};\boldsymbol{S}_i)\right]_N^\top \bar{\boldsymbol{K}} \left[\partial_{f_{\theta^{t+1}}}\mathcal{L}(f_{\theta^{t+1}};\boldsymbol{S}_i) - \partial_{f_{\theta^t}}\mathcal{L}(f_{\theta^t};\boldsymbol{S}_i)\right]_N \\
= \quad & \left[E_{\boldsymbol{S}_i}\left(\frac{\partial\mathcal{L}(f_{\theta^{t+1}})}{\partial f_{\theta^{t+1}}} - \frac{\partial\mathcal{L}(f_{\theta^t})}{\partial f_{\theta^t}}\right)\right]_N^\top \bar{\boldsymbol{K}} \left[E_{\boldsymbol{S}_i}\left(\frac{\partial\mathcal{L}(f_{\theta^{t+1}})}{\partial f_{\theta^{t+1}}} - \frac{\partial\mathcal{L}(f_{\theta^t})}{\partial f_{\theta^t}}\right)\right]_N \\
\leq \quad & \tau^2 \left[E_{\boldsymbol{S}_i}(f_{\theta^{t+1}} - f_{\theta^t})\right]_N^\top \bar{\boldsymbol{K}} \left[E_{\boldsymbol{S}_i}(f_{\theta^{t+1}} - f_{\theta^t})\right]_N \\
= \quad & \tau^2 \left\langle (f_{\theta^{t+1}} - f_{\theta^t}), [K_{\boldsymbol{S}_i}]_N^\top\right\rangle_{\mathcal{H}} \cdot \bar{\boldsymbol{K}} \cdot \left\langle [K_{\boldsymbol{S}_i}]_N, (f_{\theta^{t+1}} - f_{\theta^t})\right\rangle_{\mathcal{H}} \\
= \quad & \eta^2\tau^2 \cdot \left[\partial_{f_{\theta^t}}\mathcal{L}(f_{\theta^t};\boldsymbol{S}_i)\right]_N^\top \frac{\left\langle [K_{\boldsymbol{S}_i}]_N, [K_{\boldsymbol{S}_i}]_N^\top\right\rangle_{\mathcal{H}}}{NS} \cdot \bar{\boldsymbol{K}} \cdot \frac{\left\langle [K_{\boldsymbol{S}_i}]_N, [K_{\boldsymbol{S}_i}]_N^\top\right\rangle_{\mathcal{H}}}{NS} \cdot \left[\partial_{f_{\theta^t}}\mathcal{L}(f_{\theta^t};\boldsymbol{S}_i)\right]_N \ .
\end{aligned}
\tag{47}
$$

Given that the canonical kernel is bounded above by a constant $\gamma > 0$, we have

$$
\left\langle [K_{\boldsymbol{S}_i}]_N, [K_{\boldsymbol{S}_i}]_N^\top\right\rangle_{\mathcal{H}} \leq \gamma \left\langle \mathbf{1}_{NS}, \mathbf{1}_{NS}^\top\right\rangle,
$$

and

$$
\bar{\boldsymbol{K}} \leq \frac{\gamma}{NS} \left\langle \mathbf{1}_{NS}, \mathbf{1}_{NS}^\top\right\rangle \ .
$$

Therefore, $\Phi$ is bounded above by

$$
\begin{aligned}
\Phi \quad \leq \quad & \frac{\gamma}{NS} \left\langle \left[\partial_{f_{\theta^t}}\mathcal{L}(f_{\theta^t};\boldsymbol{S}_i)\right]_N^\top, \mathbf{1}_{NS}\right\rangle \left\langle \mathbf{1}_{NS}^\top, \left[\partial_{f_{\theta^t}}\mathcal{L}(f_{\theta^t};\boldsymbol{S}_i)\right]_N\right\rangle \\
= \quad & \frac{\gamma}{NS} \left(\sum_{i=1}^N \sum_{j=1}^S \partial_{f_{\theta^t}}\mathcal{L}(f_{\theta^t};\boldsymbol{S}_{i(j,:)})\right)^2 \ .
\end{aligned}
\tag{48}
$$

Moreover, the last term in Equation 47 is also bounded above:

$$
\begin{aligned}
& \eta^2\tau^2 \cdot \left[\partial_{f_{\theta^t}}\mathcal{L}(f_{\theta^t};\boldsymbol{S}_i)\right]_N^\top \frac{\left\langle [K_{\boldsymbol{S}_i}]_N, [K_{\boldsymbol{S}_i}]_N^\top\right\rangle_{\mathcal{H}}}{NS} \cdot \bar{\boldsymbol{K}} \cdot \frac{\left\langle [K_{\boldsymbol{S}_i}]_N, [K_{\boldsymbol{S}_i}]_N^\top\right\rangle_{\mathcal{H}}}{NS} \cdot \left[\partial_{f_{\theta^t}}\mathcal{L}(f_{\theta^t};\boldsymbol{S}_i)\right]_N \\
\leq \quad & \eta^2\tau^2 \left[\frac{\gamma}{NS}\sum_{i=1}^N\sum_{j=1}^S \partial_{f_{\theta^t}}\mathcal{L}(f_{\theta^t};\boldsymbol{S}_{i(j,:)})\right]^\top \cdot \bar{\boldsymbol{K}} \cdot \left[\frac{\gamma}{NS}\sum_{i=1}^N\sum_{j=1}^S \partial_{f_{\theta^t}}\mathcal{L}(f_{\theta^t};\boldsymbol{S}_{i(j,:)})\right]_N \\
\leq \quad & \frac{\eta^2\tau^2\gamma^3}{NS} \left\langle \left[\frac{1}{NS}\sum_{i=1}^N\sum_{j=1}^S \partial_{f_{\theta^t}}\mathcal{L}(f_{\theta^t};\boldsymbol{S}_{i(j,:)})\right]_N^\top, \mathbf{1}_{NS}\right\rangle \left\langle \mathbf{1}_{NS}^\top, \left[\frac{1}{NS}\sum_{i=1}^N\sum_{j=1}^S \partial_{f_{\theta^t}}\mathcal{L}(f_{\theta^t};\boldsymbol{S}_{i(j,:)})\right]_N\right\rangle \\
= \quad & \frac{\eta^2\tau^2\gamma^3}{NS} \left(\sum_{i=1}^N\sum_{j=1}^S \partial_{f_{\theta^t}}\mathcal{L}(f_{\theta^t};\boldsymbol{S}_{i(j,:)})\right)^2 \ .
\end{aligned}
\tag{49}
$$

Thus, by combining Equations 46, 47, 48, and 49, we get

$$
\Upsilon \leq -\eta\gamma\left(\frac{3}{4} - \eta^2\tau^2\gamma^2\right)\left(\frac{1}{NS}\sum_{i=1}^N\sum_{j=1}^S \partial_{f_{\theta^t}}\mathcal{L}(f_{\theta^t};\boldsymbol{S}_{i(j,:)})\right)^2,
\tag{50}
$$

which means

$$\frac{\partial \mathcal{L}}{\partial t} \leq \Upsilon \leq -\eta \gamma \left( \frac{3}{4} - \eta^2 \tau^2 \gamma^2 \right) \left( \frac{1}{NS} \sum_{i=1}^{N} \sum_{j=1}^{S} \partial_{f_{\theta^t}} \mathcal{L}(f_{\theta^t}; \boldsymbol{S}_{i(j,:)}) \right)^2 . \tag{51}$$

Hence, if $\eta \leq \frac{1}{2\tau\gamma}$, it follows that

$$\frac{\partial \mathcal{L}}{\partial t} \leq -\frac{\eta \gamma}{2} \left( \frac{1}{NS} \sum_{i=1}^{N} \sum_{j=1}^{S} \partial_{f_{\theta^t}} \mathcal{L}(f_{\theta^t}; \boldsymbol{S}_{i(j,:)}) \right)^2 = -\frac{\eta \gamma}{2} \left( \frac{1}{NS} \sum_{i=1}^{N} \sum_{j=1}^{S} \frac{\partial \mathcal{L} \left( f_{\theta^t}(\boldsymbol{S}_i)_{(j,:)}, \boldsymbol{y}_{i(j,:)} \right)}{\partial f_{\theta^t}(\boldsymbol{S}_i)_{(j,:)}} \right)^2 . \tag{52}$$

∎

## C    EXPERIMENT DETAILS

### C.1    LLMs TRAINING SETTING

All experiments were conducted on 4 NVIDIA A100 (80GB) GPUs. We employ LoRA fine-tuning following the Alpaca (Taori et al., 2023) and Pizza (Meng et al., 2024) implementation strategy. Specifically, we optimize with AdamW using a batch size of 128, a learning rate of $2e-5$, cosine annealing scheduling (Loshchilov & Hutter, 2017), and a warmup ratio of 0.03, without weight decay. The training objective computes loss only over responses from the selected datasets. We configure LoRA with lora_alpha=lora_r, set lora_dropout, and insert adapters into all linear layers of the base model. Both the backbone and adapters are trained in Float32 precision.

**AtteNT Setting**    In this experiment, we adopt a straightforward variant of AtteNT: the model is trained on the full dataset during the first epoch, after which only the AtteNT selected subset is used in subsequent epochs. Selection is guided by the per-sample loss scores within each epoch, effectively directing the model's attention toward harder examples. Since pretrained models already perform well on most instances, emphasizing more challenging data in later epochs is expected to yield greater fine-tuning benefits. Following prior findings in Rho-1 (Lin et al., 2024), we set the selection ratio to 70%.

**Dataset Building**    The ImageNetS50 dataset is derived from the ImageNet-1k benchmark, and its construction requires access to a local copy of ImageNet-1k. Following the official repository (Gao et al., 2021), we generate ImageNetS50 by running `data_preparation.sh` with the option `-mode=50`. Semantic segmentation annotations are obtained using `datapreparation_anno.sh`. For depth annotations, we employ the Mask2Former (Cheng et al., 2022) framework, utilizing its released code and pretrained models to generate pseudo-labels. In addition, we directly download the full NYUv2 dataset, where the official semantic segmentation and depth test sets are used for evaluation.

### C.2    ViTs TRAINING SETTING

We adopt ViT-B (Dosovitskiy et al., 2020) with a $16 \times 16$ patch size as the backbone for our MAE experiments and evaluate performance on ImageNet-S50. Training is performed using AdamW with a base learning rate of $1e-4$ and weight decay of 0.05. The learning rate is linearly warmed up for 40 epochs, followed by cosine decay scheduling (Loshchilov & Hutter, 2017). We train with a batch size of 2048 on 4 A100 GPUs, leveraging automatic mixed precision for efficiency. Data augmentation is limited to standard transformations: random cropping with scale sampled from [0.2, 1.0] and aspect ratio from [0.75, 1.33], resizing to $224 \times 224$, and random horizontal flipping with probability 0.5. A full specification of hyperparameters for pretraining and fine-tuning is provided in Tables 5 and 6.

Table 5: Hyperparameters for pre-training Multi-Modal MAE.

| Hyperparam | Baseline | AtteNT |
|---|---|---|
| Batch Size | 2048 | 2048 |
| Learning Rate | 1e-4 | 1e-4 |
| Min Learning Rate | 1e-6 | 1e-6 |
| Weight Decay | 0.05 | 0.05 |
| Adamw $\epsilon$ | 1e-8 | 1e-8 |
| Adamw $\beta_1$ | 0.9 | 0.9 |
| Adamw $\beta_2$ | 0.95 | 0.95 |
| Epoch | 800 | 800 |
| Warm up Epoch | 40 | 40 |
| Learning Rate Schedule | cosine decay | cosine decay |
| Non-masked tokens | 98 | 98 |
| Input resolution | 224×224 | 224×224 |
| Augmentation | RandomResizeCrop | RandomResizeCrop |
| Dropout | 0.0 | 0.0 |
| Patch Size | 16 | 16 |
| Selection | {None} | {Random, Hard, Soft} |

Table 6: Hyperparameters for fine-tuning Multi-Modal MAE on various downtasks. The augmentation strategy LSJ is large scale jittering (Ghiasi et al., 2021). We use drop path (Huang et al., 2016) in classification and semantic segmentation tasks.

| Hyperparam | ImageNetS50 | NYUv2(S) | NYUv2(D) |
|---|---|---|---|
| Epoch | 100 | 100 | 2000 |
| Warm up Epoch | 5 | 20 | 100 |
| Batch Size | 1024 | 1024 | 2048 |
| Learning Rate | 4e-3 | 1e-4 | 1e-4 |
| Min Learning Rate | 1e-6 | 1e-6 | 0 |
| Weight Decay | 0.05 | 0.05 | 1e-4 |
| Adamw $\beta_1$ | 0.9 | 0.9 | 0.9 |
| Adamw $\beta_2$ | 0.999 | 0.999 | 0.999 |
| Layer Decay | 0.65 | 0.75 | 0.75 |
| Patch Size | 16 | 16 | 16 |
| Drop path | 0.1 | 0.1 | / |
| LR Schedule | cosine decay | cosine decay | cosine decay |
| Input resolution | 224×224 | 224×224 | 256×256 |
| Augmentation | Rand(9, 0.5) | LSJ | LSJ |

**AtteNT Setting** We employ an enhanced AtteNT strategy that dynamically selects training data based on per-sample loss scores. Specifically, data selection in the first epoch is guided by each sample's initial loss, and the selection is periodically updated by recomputing loss scores after fixed intervals. This updated subset is then used to initiate the next training stage. Moreover, we incorporate a dynamic selection ratio from 20% to 80%, following the adaptive scheme proposed by (Zhang et al., 2023b). As demonstrated in Section 5, this approach achieves a favorable trade-off between efficiency and performance.

**Dataset Building** All datasets used in our experiments are available on HuggingFace:

- Mathematical Reasoning: Training on `meta-math/MetaMathQA`; evaluation on `openai/gsm8k` and `hendrycks/MATH`.
- Code Generation: Training on `m-a-p/CodeFeedback` (restricted to Python samples); evaluation on `openai/humaneval` and `google/mbpp`.
- Multi-Turn Dialogue: Training on `WizardLM/evol_instruct_196k`; evaluation on `lmsys/mt-bench`.

# D ADDITIONAL EXPERIMENTS

## D.1 ABLATION OF SAMPLE RATIO

Table 7: Sample Ratio Study of ViT models.

| Ratio / Tasks | 100 | 90 | 80 | 70 | 60 | 50 | 40 |
|---|---|---|---|---|---|---|---|
| ImageNetS50 | 92.2 | 91.8 | 91.4 | 85.6 | 78.8 | 64.5 | 58.2 |
| NYUv2(S) | 51.9 | 51.1 | 50.2 | 46.8 | 38.2 | 29.1 | 21.9 |
| NYUv2(D) | 52.1 | 52.2 | 51.6 | 48.3 | 42.4 | 36.3 | 30.5 |

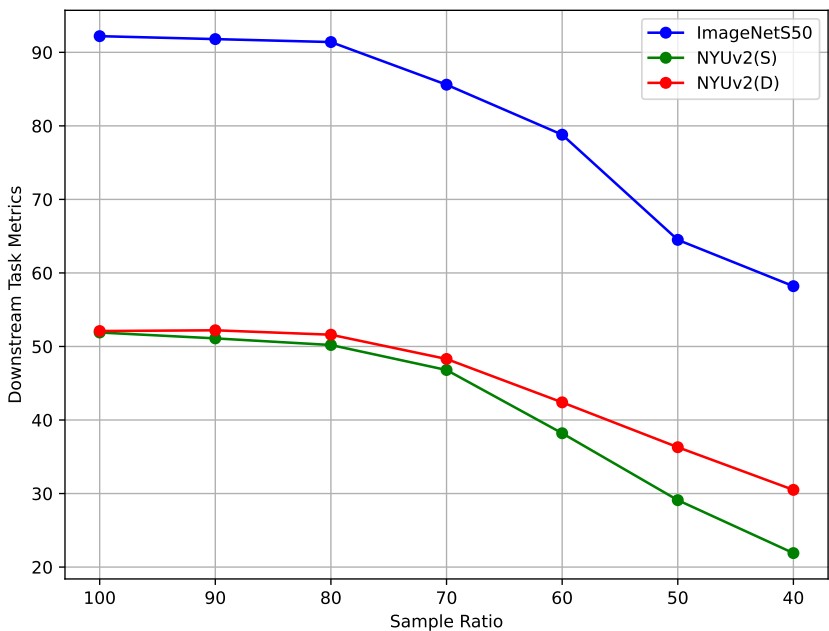

Figure 4: Downstream Task Performance vs Sample Ratio.

In this section, we investigate the impact of the AtteNT algorithm's sample ratio on downstream tasks. In Table 7, we compare the results based on the ViT model using different fixed sample ratios.

As we can see from Fig 4, there is a noticeable drop in performance around the 80% selection ratio. This suggests that, during training, a portion of the data remains relatively unchanged and contributes less to performance when its selection ratio falls below a certain threshold (Lin et al., 2024; Katharopoulos & Fleuret, 2018). When the ratio of unselected samples during training is lower than this threshold, the model can maintain its performance. Interestingly, a small data drop can even act as a form of noise reduction.

## D.2 COMPARISON TO ESTABLISHED METHODS

Table 8: Performance Comparison of Different Methods.

| Methods | Time(↓) | ImageNetS50(↑) | NYUv2(S)(↑) | NYUv2(D)(↑) |
|---|---|---|---|---|
| AtteNT(Ours) | **980m** | **92.3** | **52.6** | **57.2** |
| Class Weight Sampling | 1108m | 90.4 | 48.2 | 52.0 |
| Fixed Weight Sampling | 1065m | 89.6 | 49.7 | 54.6 |
| GradNorm Sampling (Chen et al., 2018) | 1112m | 91.9 | 52.4 | 55.8 |

As shown in Table 8, we also performed experiments comparing AtteNT with three simple yet representative sample-selection baselines to compare with traditional greedy algorithm: The first method, Class-Weight Sampling, assigns sampling weights inversely proportional to the number of samples per class, aiming to encourage the model to treat all classes more equally (sampling weight = 1 / class frequency). The second, Fixed-Weight Sampling, assigns fixed sampling ratios based on prior beliefs about task difficulty. In our setting, we consider the classification task easier than semantic segmentation and depth estimation, so we reduce the sampling rate for the RGB modality and set fixed sampling weights to 1 : 2 : 2 (RGB : SemSeg : Depth). The third method, GradNorm Sampling (Chen et al., 2018), dynamically adjusts the sampling weights for data groups (RGB, SemSeg, Depth) based on their gradient contributions during training. All methods were trained for 800 epochs, with the total sampling budget fixed at 70% for each baseline.

Across all comparisons, AtteNT consistently achieves higher efficiency and stronger predictive performance. These results indicate that AtteNT's gains do not arise from generic greedy sampling heuristics, but from its principled nonparametric teaching mechanism, which adapts to model uncertainty and task interactions more effectively than existing selection strategies.

### D.3    VISUALIZING NTK ANALYSIS

To empirically confirm that the neural tangent kernel quickly stabilizes in real vision transformer training, we track the NTK on 10 fixed training points during an 800-epoch run of the Multi-Modal MAE backbone:

- Figure 5: Frobenius norm of the difference between the empirical NTK at epoch and the canonical kernel. The difference falls sharply within the first 50 epochs and stays near zero thereafter.

- Figure 6: Heatmaps of the $10 \times 10$ NTK at selected checkpoints. A clear pattern is already visible at epoch 139, and from epoch 219 onward the heatmaps are virtually identical and remain unchanged through epoch 799.

These quantitative and qualitative results jointly show that the empirical NTK converges extremely rapidly (within 50–200 epochs) and remains close to the canonical kernel for the rest of training.

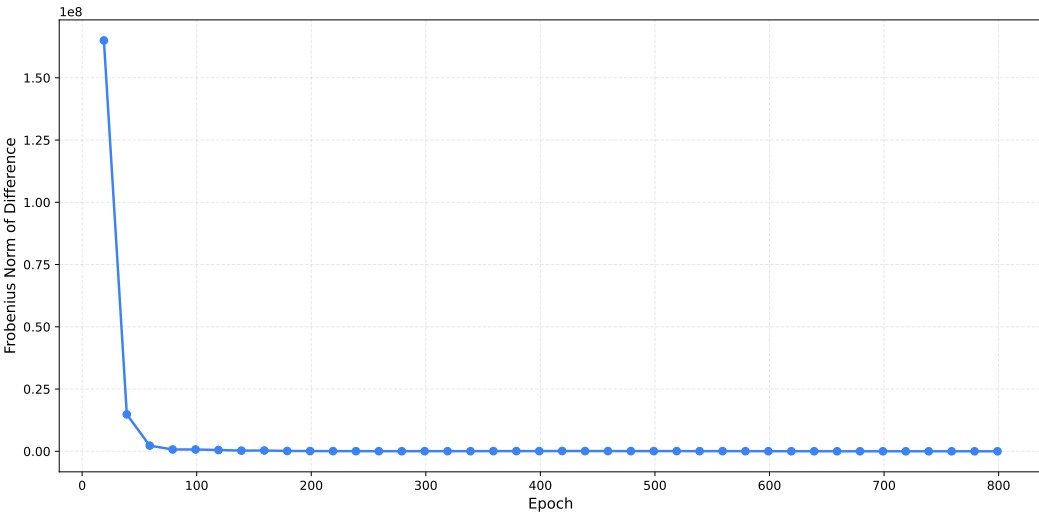

Figure 5: Frobenius norm of the difference between the empirical NTK at different training steps and the canonical kernel.

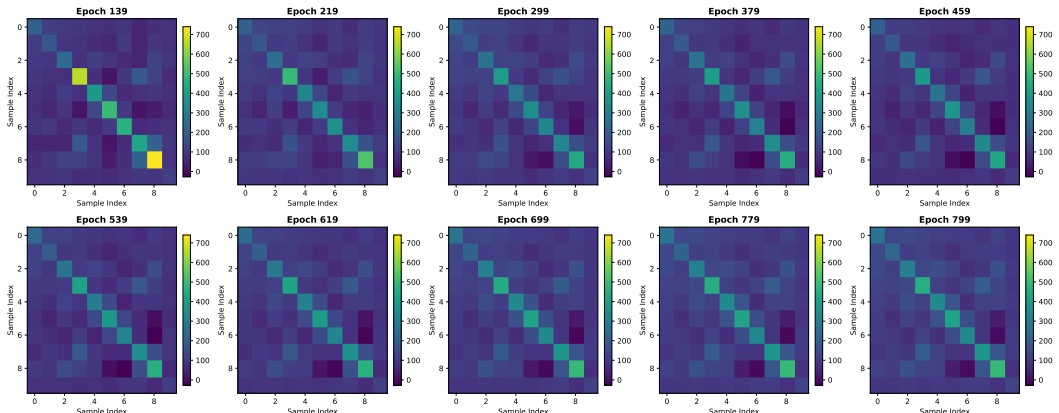

Figure 6: Evolution of the empirical $10 \times 10$ NTK matrix during training. Color represents value $K_{\theta^t}(\boldsymbol{S}_i, \boldsymbol{S}_j)$. The matrix stabilizes visually after 200 epochs and shows negligible changes thereafter.

# E    THE USE OF LARGE LANGUAGE MODELS (LLMS)

We used large language models for language polishing, such as grammar and phrasing. And we also use AI to assist with code completion. All research ideas, methods, analyses, figures, tables, and conclusions were solely developed by the authors. The authors take full responsibility for all content.

