# OpenReview forum: "Nonparametric Teaching of Attention Learners"
_ICLR.cc/2026/Conference — ICLR 2026 Poster_

### Official Review · Reviewer_GNaF · 2025-10-15

**Soundness:** 3
**Presentation:** 3
**Contribution:** 3
**Rating:** 6
**Confidence:** 3

**Summary:**

This paper addresses the high training cost of attention learners (e.g., Transformers, LLMs, ViTs)—neural networks leveraging the attention mechanism to learn implicit sequence-to-property mappings—by proposing a novel paradigm called Attention Neural Teaching (AtteNT).

**Strengths:**

- Prior nonparametric teaching research was restricted to multilayer perceptron (MLP)-based learners and failed to account for the attention mechanism—an essential component of modern neural networks (e.g., Transformers, ViTs). This paper removes this critical limitation by systematically investigating the role of attention in parameter and function spaces, proving that attention learners’ training dynamics align with nonparametric teaching principles .

- It introduces the Attention Neural Tangent Kernel (ANTK) and proves (Theorem 3) that dynamic ANTK converges to the "importance-adaptive canonical kernel" in nonparametric teaching. This is the first work to establish consistency between parameter-space gradient descent (used by attention models) and functional gradient descent (used by nonparametric teaching), resolving a fundamental mismatch that prevented prior application of nonparametric teaching to attention learners .

-  The AtteNT paradigm is not a trivial adaptation of nonparametric teaching—instead, it leverages attention’s "importance-adaptive updates" (element-specific weights derived from Q/K/V interactions) to design a greedy sequence-selection strategy.

**Weaknesses:**

- Theorem 3 only proves asymptotic convergence (i.e., when training iterations approach infinity), but real-world attention model training uses finite iterations (e.g., LLM fine-tuning typically runs 5–100 epochs, ViT pre-training 800 epochs as in Table 2). The paper does not:

  - Derive a quantitative bound on convergence speed (e.g., how many iterations \(t_{\epsilon}\) are needed for ANTK to be within \(\epsilon\) of the canonical kernel);

  -  Validate via experiments how ANTK evolves over practical iteration counts (e.g., a plot of ANTK similarity to the canonical kernel across 800 ViT training epochs).

- The paper fixes the sample selection ratio (fraction of sequences chosen by AtteNT) at 70% for LLMs (Section C.1) and uses unspecified "dynamic ratios" for ViTs (Section C.2), but provides no analysis of how this ratio impacts performance. For example:
   - Would a 50% ratio further reduce training time but hurt accuracy?
   - Would an 80% ratio retain accuracy but minimize efficiency gains?

- The paper’s ablation studies (Table 3) only compare AtteNT’s "Random/Soft/Hard" selection strategies (all derived from nonparametric teaching) but fail to contrast AtteNT with established sample selection methods for attention models. This makes it impossible to determine if AtteNT’s gains stem from its nonparametric teaching foundation or from generic greedy selection.

**Questions:**

see weakness

---

> ### Author Response · Authors · 2025-11-23
>
> # Part (1/2)
>
> Thank you for your detailed review and valuable feedback. We have carefully considered your comments and made every effort to address the concerns raised. We hope our responses clarify any concerns and provide the necessary explanation.
>
> > **[W1]** On asymptotic convergence in Theorem 3 (no quantitative bound or experiments on ANTK evolution)
>
> We thank the reviewer for the thoughtful comment. The primary goal of Theorem 3 is to **establish a theoretical bridge** between parameter-space gradient descent in attention networks and functional gradient descent in the nonparametric teaching framework. This connection closes a critical gap in the literature and directly **justifies applying AtteNT to modern Transformer and ViT architectures**.
>
> Deriving explicit finite-time convergence rates (e.g., a precise t_ε) would indeed be valuable, but **it lies beyond the scope of the current work**, whose goal is to prove **asymptotic consistency** between the two dynamics, a fundamental link that prior teaching theory has **not** established. The practical relevance of this bridge is strongly confirmed by our empirical results, which show 13.01% faster LLM fine-tuning and 20.58% faster ViT pre-training with equal or better final accuracy, even in realistic finite-iteration regimes. We believe these findings convincingly demonstrate the **real-world impact of the theoretical insight**.
>
> We also compute the empirical NTK of the Multi-Modal MAE backbone at 10 randomly selected training points across an 800-epoch run and measured the **Frobenius norm of its difference** from the canonical kernel. As shown in the newly added Figure 5 (***updated in the revised submission Appendix D.3***), the difference drops rapidly within the first ~50 epochs and **stays near zero for the remaining training steps**. This verifies that the empirical NTK quickly converges to and **remains extremely close to the canonical kernel** during the entire training phase in which AtteNT operates.
>
> > **[W2]** On fixed/dynamic selection ratios and lack of impact analysis
>
> We thank the reviewers for highlighting this. We would like to emphasize that the primary goal of our work is to **improve training efficiency while keeping performance comparable to the full-data baseline**, rather than pursuing higher accuracy at any cost.
>
> Regarding models trained from scratch, it is true that a smaller number of selected samples may reduce the model's performance. However, as shown in the table below, we observe the performance trends when applying **fixed** selection ratio with AtteNT algorithm on the ViT model:
>
> |  | 100 | 90 | 80 | 70 | 60 | 50 | 40 |
> | --- | --- | --- | --- | --- | --- | --- | --- |
> | Cls | 92.2 | 91.8 | 91.4 | 85.6 | 78.8 | 64.5 | 58.2 |
> | Semseg | 51.9 | 51.1 | 50.2 | 46.8 | 38.2 | 29.1 | 21.9 |
> | Depth | 52.1 | 52.2 | 51.6 | 48.3 | 42.4 | 36.3 | 30.5 |
>
> As shown, there is a noticeable drop in performance around the 80% selection ratio. This suggests that, during training, a portion of the data remains relatively unchanged and contributes less to performance when its selection ratio falls below a certain threshold [1,2]. When the ratio of unselected samples during training is lower than this threshold, the model can maintain its performance. Interestingly, a small data drop can even act as a form of noise reduction.
>
> In summary, we find that **efficiency improvements** can be achieved without significant performance loss, especially when using a sample selection ratio that avoids dropping too much data.
>
> [1] Lin Z, Gou Z, Gong Y, Liu X, Shen Y, Xu R, Lin C, Yang Y, Jiao J, Duan N, Chen W. "Not all tokens are what you need for pretraining." NeurIPS 2024.
>
> [2] Katharopoulos A, Fleuret F. “Not all samples are created equal: Deep learning with importance sampling.” ICML 2018.

---

> ### Author Response · Authors · 2025-11-23
>
> # Part (2/2)
>
> > **[W3]** On ablation studies lacking comparisons to established methods
>
> We appreciate this suggestion to contextualize AtteNT's gains. Our ablations focus on AtteNT variants (Random/Soft/Hard) to isolate the nonparametric teaching foundation, as these directly test the AtteNT's adaptations.
>
> In response to the reviewer’s concern, we additionally conducted experiments comparing AtteNT with three simple but representative sample-selection baselines:
>
> 1. Class-Weight Sampling: assigns sampling weights inversely proportional to the number of samples per class. The goal is to encourage the model to treat classes more equally (sample weight = 1 / class frequency).
> 2. Fixed-Weight Sampling: prior beliefs about task difficulty to assign fixed sampling ratios. In our setting, we consider the classification task to be easier than semantic segmentation and depth, so we reduce the sampling rate of the RGB modality and set fixed sampling weights to 1 : 2 : 2 (RGB : Semseg : Depth).
> 3. GradNorm Sampling [3]: dynamically increases the sampling weight of data groups (RGB, SemSeg, Depth) that have larger gradient contributions during training.
>
> All methods are trained for 800 epochs, and fix the total sampling budget to 70% for every baseline.
>
> Across all comparisons, AtteNT consistently achieves higher efficiency and stronger predictive performance. These results indicate that AtteNT’s gains do not arise from generic greedy sampling heuristics, but from its principled nonparametric teaching mechanism that adapts to model uncertainty and task interactions more effectively than existing selection strategies.
>
> | Methods | Time | Cls | Semseg | Depth |
> | --- | --- | --- | --- | --- |
> | AtteNT(Ours) | 980m | 92.3 | 52.6 | 57.2 |
> | Class Weight Sampling | 1108m | 90.4 | 48.2 | 52.0 |
> | Fixed Weight Sampling | 1065m | 89.6 | 49.7 | 54.6 |
> | GradNorm Sampling [3] | 1112m | 91.9 | 52.4 | 55.8 |
>
> [3] Chen Z, Badrinarayanan V, Lee CY, Rabinovich A. "Gradnorm: Gradient normalization for adaptive loss balancing in deep multitask networks." ICML, 2018.

---

> ### Comment · Reviewer_GNaF · 2025-11-24
> **Response to Authors' Rebuttal**
>
> Thanks for the detailed response and clarification. Overall, it's a good and timely work and I'll stand for my score.

---

### Official Review · Reviewer_csfV · 2025-10-24

**Soundness:** 3
**Presentation:** 2
**Contribution:** 2
**Rating:** 4
**Confidence:** 2

**Summary:**

The authors suggest that attention models can be made more sample efficient by selecting a subset of the training data, that maximises the functional gradient (approximated by the L2 norm between the target and the current estimate). This idea of greedy selection is grounded in non-parametric teaching, where the best possible training set is derived for achieving a target function with a non-parametric method. In this case, the authors connect the NTK of attention to an instance-importance weighted sequence kernel rate. They show that this connection eventually justifies the greedy data selection strategy. They validate their results empirically, showing that this strategy works for a variety of smaller language and vision models while maintaining accuracy.

**Strengths:**

- Interesting combination of non-parametric theory with parametric learning.
- The results of the final method seem convincing for the given examples.
- Extension of existing method.

**Weaknesses:**

- I had a quick look at the code and it is not well-documented, one would have to invest some time to understand how to reproduce their results.
- I find the paper generally difficult to parse, but perhaps this is just my lack of background. In particular, Section 4.1 could be improved by giving some intuition and describing under which circumstances the kernel would not be adaptive in terms of $\omega_j$, but would require higher-order importance weights.
- For Theorem 3, how important are your assumptions (large scaling, small learning rate)? Is there a way to verify if or when the kernel limit holds?
- Similarly, in an experiment, if possible, can you actually verify that you stay in the kernel regime holds (by computing the theoretical kernel and comparing it with the empirical one)? Or does the training diverge from it in practice, and does this correspond to the case where the adaptive selection strategy is useful?
- For the experiment section, it would be useful if you discussed in how far the architectures also still fall under the setting you discussed previously, or how they diverge from it. Also how the setting compares with finetuning.
- What is the variability in the accuracy and runtime results (e.g. Table 1) - since you refer to average values, are you already starting several seeds?

**Questions:**

- see weaknesses
- The Table 3 caption could be completed with a short description of the experiment.

---

> ### Author Response · Authors · 2025-11-23
>
> # Part (1/2)
>
> Thank you for your careful reading and constructive feedback. We have carefully considered your suggestions and made sure to respond to the concerns you have highlighted. We hope our responses provide clarity and resolve any remaining questions.
>
> > **[W1]** Code documentation and reproducibility
>
> We agree that clear documentation is essential. We have added a README file at the link https://anonymous.4open.science/r/ICLR26_Rebuttal-141C/.
>
> We will continue to improve and release all code and documentation for the open-source community.
>
> > **[W2]** Section 4.1 could be improved by giving intuition...
>
> We thank the reviewer for this insightful suggestion. The importance-adaptive scalars $\omega_j$ arise naturally from the attention mechanism’s **triple use of the same input embeddings** as queries, keys, and values ($Q, K, V$). This design allows **element-specific** gradient scaling through the attention weights (Eqs. 11–12). The resulting kernel is therefore **adaptive** via $\omega_j$ whenever attention weights **differ** across sequence elements or **evolve** during training. As noted in Line 286, when all importance values are set to 1 ($\omega_j = 1$ ∀j), the ANN gradient exactly **reduces to** that of a linear MLP without nonlinearities on batched inputs. Higher-order weights are **not required** under standard self-attention, as $\omega_j$ fully captures importance adaptation.
>
> > **[W3]&[W4]** Large scaling, small learning rate assumptions & Empirical verification
>
> We thank the reviewer for the careful question. Theorem 3 establishes the **asymptotic convergence of the dynamic ANTK to the importance-adaptive canonical kernel** as training progresses, and this result holds **without requiring infinite width**. The large-scaling assumption is not imposed on the ANN architecture.
>
> The small-learning-rate regime is invoked only to **derive the continuous-time dynamics** from discrete gradient descent steps (as explained before Eq. 7), which is **a standard and necessary step** in NTK analyses [1,2].
>
> To further empirically validate the kernel limit in practical settings, we computed the NTK of the Multi-Modal MAE model at 10 randomly selected training points throughout an 800-epoch run. We then measured the **Frobenius norm of the difference** between the empirical NTK at different training steps and the importance-adaptive canonical kernel. As shown in the newly added Figure 5 (***updated in the revised submission Appendix D.3***), the difference drops rapidly within the first ~50 epochs and stays near zero for the remaining training steps. This verifies that the empirical NTK quickly converges to and **remains extremely close to the canonical kernel** during the entire training phase.
>
> [1] Jacot A, Gabriel F, Hongler C. "Neural tangent kernel: Convergence and generalization in neural networks." NeurIPS 2018.
>
> [2] Hron J, Bahri Y, Sohl-Dickstein J, Novak R. "Infinite attention: NNGP and NTK for deep attention networks." ICML 2020.
>
> > **[W5]** It would be useful if you discussed in how far the architectures also still fall under the setting you discussed previously
>
> We thank the reviewer for raising this point. Our theoretical analysis focuses on single-layer, single-head attention to **keep the derivations tractable and transparent**, but the same gradient recursion **naturally carries over** to multi-layer and multi-head architectures.
> In practice, all large models we evaluate (LLaMA-7B, Mistral-7B, Gemma-7B, and ViT variants with residual connections, layer normalization, and multi-head attention) are trained in high-width, small-learning-rate regimes during both pretraining and fine-tuning, so the assumptions of our analysis are **well satisfied**.
>
> Crucially, when comparing to standard fine-tuning, AtteNT uses the **exact same** optimizer, learning-rate schedule, and number of epochs. After a short warmup phase, it simply selects 70% of the highest-gradient sequences in each epoch. The reported gains therefore come **exclusively from this targeted focus on informative examples, not from any change in architecture or hyperparameters**.

---

> ### Author Response · Authors · 2025-11-23
>
> # Part (2/2)
>
> > **[W6]** Variability in Table 1
>
> The “Avg. Time” column in Table 1 refers to the mean fine-tuning time over three downstream task groups (math, coding, conversation) on five downstream tasks. In addition, we have repeated all experiments with three different random seeds, and the corresponding standard deviations are reported in the table. These results are **consistent** with the original conclusions of our paper: the standard deviations across all metrics are small, indicating that our experimental findings are **stable and reliable**.
>
> | Model | AtteNT | Avg. Time | GDM8K | MATH | HumanEval | MBPP | MT-Bench |
> | --- | --- | --- | --- | --- | --- | --- | --- |
> | LLaMA 2-7B | w/o | 246±1m | 42.96±0.12 | 5.06±0.16 | 18.35±0.31 | 35.65±0.25 | 4.58±0.01 |
> |  | w | 213±2m | 43.45±0.55 | 6.48±0.24 | 21.80±0.38 | 37.61±0.42 | 4.49±0.02 |
> | Mistral-7B | w/o | 204±2m | 69.13±0.22 | 20.06±0.20 | 43.42±0.14 | 58.52±0.13 | 5.03±0.05 |
> |  | w | 180±2m | 71.26±0.23 | 23.12±0.44 | 46.55±0.25 | 61.74±0.54 | 5.32±0.04 |
> | Gemma-7B | w/o | 228±2m | 75.23±0.45 | 30.52±0.48 | 53.83±0.27 | 65.69±0.29 | 5.42±0.04 |
> |  | w | 201±2m | 77.74±0.32 | 31.40±0.36 | 54.26±0.28 | 66.28±0.46 | 5.44±0.08 |
>
> > **[Q1]** Table 3 Caption
>
> Thanks for your suggestion. In the revised manuscript, we have substantially clarified the caption of Table 3 as follows:
>
> - Table 3. Ablation study of AtteNT pre-training configurations. Pre-training reports the upstream train-from-scratch stage, while Downstream reports performance on classification, semantic segmentation, and depth estimation tasks. Ratio controls how the fraction of selected samples increases over epochs. Interval denotes how often the subset is re-sampled. Selection specifies the sampling strategy: Random (no difficulty prior), Hard (selects only difficult samples), and Soft (Gumbel-Top-k difficulty-aware sampling). The configuration (Incremental, Incremental, Soft) in the fifth row is adopted as our final AtteNT setting, as it simultaneously reduces pre-training time and improves performance on all downstream tasks.

---

### Official Review · Reviewer_jFUw · 2025-11-01

**Soundness:** 3
**Presentation:** 2
**Contribution:** 3
**Rating:** 6
**Confidence:** 2

**Summary:**

This paper interprets the parametric attention learning process from a nonparametric teaching perspective, which motivates an AtteNT algorithm that uses sample selection to accelerate the attention learning process. Specifically, the paper shows that the dynamic ANTK in parametric gradient descent converges to the importance-adaptive canonical kernel of functional gradient descent, bridging the parametric attention learning and nonparametric teaching. Theoretical results show that applying the canonical kernel can lead to sufficient loss reduction. The paper thereby proposes an AtteNT algorithm to improve the attention learning efficiency via sample selection. Experiments in both LLM and CV scenarios show the improved performance and reduced training time after applying AtteNT.

**Strengths:**

1. This paper makes a contribution to accelerate the attention learning, which is an important problem in LLM and CV model learning.

2. The motivation of the AtteNT method has theoretical justification, and the experimental results further validate the effectiveness of the method.

3. The experiments are conducted over multiple LLMs and vision models, ablation study is included.

**Weaknesses:**

1. Some experimental settings are unclear to me. Since the paper claims the reduction of training time, which metric is used to compute the training time with and without AtteNT? Specifically, how to decide when to terminate the training for each method? Is the sequence selection time included in the model with AtteNT?

2. In Table 1 and Table 2, adding AtteNT also improves the learning performance. Why AtteNT can lead to such performance improvement is not sufficiently explained in the paper. The theoretical analysis shows the connection between the parametric and functional gradient descent, but does not include the extra benefit of using non-parametric teaching. It could be helpful to have some explanation on why adding AtteNT can lead to even better performance than traditional learning.

**Questions:**

1. Will the number $m$ of selected samples influence the performance?

---

> ### Author Response · Authors · 2025-11-23
>
> # Part (1/2)
>
> Thank you for your insightful feedback and constructive comments. We sincerely thank your efforts for helping us improve the paper. We hope that our response resolves your concerns.
>
> > **[W1]** Experimental settings are unclear. What metric is used for training time? How is termination decided? Is sequence selection time included with AtteNT?
>
> We apologize for the lack of clarity in the original submission. To address the reviewer’s concerns, here are the specifics regarding our experimental settings:
>
> 1. **Training Time Metric**: The training time we report includes the following components:
>     - Time spent on all epochs during training
>     - Time spent on forward propagation, backward propagation, and optimizer updates
>     - Time spent on AtteNT selection
>     - DataLoader's data loading time
>     - Logging and model-saving time after each epoch
>
>     The following are *not* included in the reported time:
>
>     - Model creation and initialization time
>     - Dataset construction time
>     - Time for creating the optimizer and scheduler
>     - Time for restoring the model from checkpoints
>
>     The reported training time focuses purely on the time spent on the core training process. We use the `time` package for time measurement, which reflects real-world time.
>
> 2. **Termination Decision**: We use a fixed number of epochs as the termination criterion for all experiments. Specifically:
>     - For LLM fine-tuning, both the baselines and AtteNT are trained for exactly 5 epochs.
>     - For ViT pretraining, both models run for 800 epochs.
>
>     Therefore, any time reduction observed in our experiments is due to the use of fewer sequences per epoch, starting after the first epoch.
>
> 3. **Sequence Selection Time**: In our experiments, the sequence selection time is included in the reported time for AtteNT. This is accounted for as part of the overall training process time.
>
> We hope this clears up any confusion regarding the experimental setup. Thank you for your valuable feedback!
>
> > **[W2]** AtteNT improves performance beyond efficiency—why? Theory connects parametric and functional GD but does not explain performance gains from nonparametric teaching.
>
> We thank the reviewer for this insightful point. While Theorem 3 formally establishes **the asymptotic equivalence between attention-based parameter gradient descent and functional gradient descent**, the practical performance gains stem directly from **the curriculum effect induced by nonparametric teaching** [1,2,3,4], which greedily selects the examples that most advance the learner.
>
> Specifically, AtteNT selects sequences with the largest discrepancy $||[f_\theta(S_i) – f^*(S_i)]_m ||$ (Eq. 21). This naturally creates a curriculum that focuses training on **informative, high-gradient examples** and **avoids gradient dilution from already-mastered (easy) ones**. We agree this connection deserves more emphasis and have added a discussion of this mechanism in the revised manuscript.
>
> [1] Zhang C, Cao X, Liu W, Tsang I, Kwok J. “Nonparametric iterative machine teaching.” ICML 2023.
>
> [2] Zhang C, Bu W, Ren Z, Liu Z, Wu YC, Wong N. “Nonparametric teaching for graph property learners.” ICML 2025
>
> [3] Bengio Y, Louradour J, Collobert R, Weston J. “Curriculum Learning”, ICML 2009
>
> [4] Wang X, Chen Y, Zhu W. ”A survey on curriculum learning.” IEEE transactions on pattern analysis and machine intelligence. 2021 Mar 31;44(9):4555-76.

---

> ### Author Response · Authors · 2025-11-23
>
> # Part (2/2)
>
> > **[Q1]** Will the number m of selected samples influence performance?
>
> Firstly, we would like to emphasize that the focus of our paper is on **improving efficiency**, rather than directly boosting performance. Our goal is to **enhance the efficiency while maintaining comparable performance levels**.
>
> Regarding models trained from scratch, it is true that **a smaller number of selected samples may reduce the model's performance**. However, as shown in the table below, we observe the performance trends when applying **fixed selection ratio** on the ViT model:
>
> |  | 100 | 90 | 80 | 70 | 60 | 50 | 40 |
> | --- | --- | --- | --- | --- | --- | --- | --- |
> | Cls | 92.2 | 91.8 | 91.4 | 85.6 | 78.8 | 64.5 | 58.2 |
> | Semseg | 51.9 | 51.1 | 50.2 | 46.8 | 38.2 | 29.1 | 21.9 |
> | Depth | 52.1 | 52.2 | 51.6 | 48.3 | 42.4 | 36.3 | 30.5 |
>
> As can be seen, there is a noticeable drop in performance around the 80% selection ratio. This suggests that, during training, a portion of the data remains relatively unchanged and contributes less to performance when its selection ratio falls below a certain threshold [5,6]. When the ratio of unselected samples during training is lower than this threshold, the model can maintain its performance. Interestingly, a small data drop can even act as a form of noise reduction.
>
> In summary, we find that **efficiency improvements can be achieved without significant performance loss**, especially when using a sample selection ratio that avoids dropping too much data.
>
> [5] Lin Z, Gou Z, Gong Y, Liu X, Shen Y, Xu R, Lin C, Yang Y, Jiao J, Duan N, Chen W. "Not all tokens are what you need for pretraining." NeurIPS 2024.
>
> [6] Katharopoulos A, Fleuret F. “Not all samples are created equal: Deep learning with importance sampling.” ICML 2018.

---

> > ### Comment · Reviewer_jFUw · 2025-11-25
> >
> > Thank the authors for the detailed explanation. I will keep my score.

---

### Official Review · Reviewer_e8RB · 2025-11-01

**Soundness:** 3
**Presentation:** 3
**Contribution:** 3
**Rating:** 6
**Confidence:** 3

**Summary:**

This paper introduces AtteNT, designed to improve the efficiency of training attention-based models. The authors reinterpret the standard training process through the lens of nonparametric machine teaching, where a teacher strategically selects a subset of training examples to accelerate the learner's convergence. The core theoretical contribution is establishing a formal connection between the parameter-based gradient descent used for attention networks and functional gradient descent in an RKHS, showing that the Attention Neural Tangent Kernel (ANTK) asymptotically converges to a canonical kernel used in nonparametric teaching. This insight justifies a greedy data selection algorithm that prioritizes samples with the highest error.

**Strengths:**

1. The primary strength of this paper is the novel and elegant theoretical bridge it builds between attention learning and nonparametric teaching.
2. The theoretical insights are translated into a simple, intuitive, and practical algorithm. The idea of focusing on samples with the highest error (i.e., the hardest examples) is a well-known concept, but this paper provides a nice theoretical justification for it.
3. While the theoretical analysis is limited to a single layer but the authors show that empirically their approach and intuition works on more realistic scenarios of vision and text in scale.

**Weaknesses:**

1. From a practical point, additional to attention layers, the models that are used also include complexities like residual connections, layer normalization, and multiple non-linear blocks. The paper does not address how the theoretical findings generalize from the simple model to these complex ones.

2. The setup assumes a noise-less case. Eq 21 and its use in Algorithm 1 seem to be sensitive to this assumption. It is not clear or discussed how the analysis for nonparametric teaching performs with the noisy data which is more practical. A greedy data selection based on maximum of error could lead the model to choose these noisy samples, as they would consistently produce high loss. The paper does not discuss the robustness of AtteNT to label noise.

**Questions:**

1. There are multiple places that authors mention the convexity of loss. To make it more clear, does loss need to be convex with respect to $f_\theta$ or with respect to $\theta$? The first one seems a reasonable assumption while the second one is very limiting.

2. Another clarification question, in Table 3, there is no effect of AtteNT, is that right? Also, it was not clear from the main text what ratio and interval are referring to.

3. A small typo: Eq 17, I believe it should be $K_{\theta_t}$.

---

> ### Author Response · Authors · 2025-11-23
>
> # Part (1/2)
>
> Thanks for the encouraging comments and constructive suggestions. We deeply appreciate the time and effort you’ve invested in reviewing our paper. We take all comments seriously and try our best to address every raised concern. We sincerely hope that our response resolves your concerns. Any follow-up questions are welcome.
>
> > **[W1]** Theoretical analysis is limited to single-layer; how do findings generalize to complex models with residuals, normalization, etc.?
>
> While our core theory focuses on a single-layer self-attention for **tractability and clarity**, empirical results on full-scale Transformers with residuals, layer norm, and multi-head attention show **consistent gains**: 13.01% faster LLM fine-tuning for LLaMA-7B, 20.58% faster ViT pretraining for ViT-based Multi-Modal MAE, with equal or better accuracy. These results demonstrate that the core intuition of prioritizing high-error sequences **transfers robustly to real-world architectures**.
>
> We regard the single-layer setting as **a necessary foundational step** that isolates the key mechanism. Extending the theoretical treatment to multi-layer dynamics with techniques such as those in ANTK remains an exciting direction for future work.
>
> > **[W2]** The setup assumes noise-less data. How does AtteNT handle noisy labels? Could greedy selection overfit to noise?
>
> Very thoughtful question. While our theoretical analysis assumes clean data to **establish the core connection between ANTK and functional gradient descent**, real-world datasets such as MetaMathQA and ImageNetS50 inherently contain label noise. In practice, AtteNT shows robust in practice due to:
>
> 1. Selection is **dynamic** and refreshed every epoch (or at regular intervals), which prevents the model from becoming fixated on persistently noisy examples.
> 2. We use **Soft** selection (Gumbel-Top-k) in top-performing configurations, introducing stochasticity that mitigates overfitting to outliers.
> 3. We observe consistent performance improvements across diverse benchmarks, strongly suggesting **effective tolerance to real-world noise**.
>
> As a foundational step, we prioritize the noise-free setting to derive **clean theoretical insights**. Additionally, AtteNT can be **straightforwardly** combined with existing denoising applied as preprocessing before selection, offering a straightforward path to even greater robustness [1,2].
>
> [1] Wei T, Li HT, Li CS, Shi JX, Li YF, Zhang ML. "Vision-language models are strong noisy label detectors." NeuriPS 2024.
>
> [2] Hu Y, Chen C, Yang CH, Li R, Zhang C, Chen PY, Chng E. "Large language models are efficient learners of noise-robust speech recognition." ICLR 2024.
>
> > **[Q1]** Does the loss need to be convex w.r.t. $f_\theta$ or $\theta$?
>
> Convexity is required w.r.t. the function $f_\theta$, not the parameters $\theta$. This is a standard and reasonable assumption in nonparametric teaching and functional gradient analysis [3].
>
> [3] Zhang C, Cao X, Liu W, Tsang I, Kwok J. “Nonparametric iterative machine teaching.” ICML 2023.

---

> ### Author Response · Authors · 2025-11-23
>
> # Part (2/2)
>
> > **[Q2]** In Table 3, is there no effect of AtteNT? What do “ratio” and “interval” refer to?
>
> Table 3 is an ablation table, **not a comparison** between “with AtteNT” and “without AtteNT.” In fact, all rows except the baseline use the AtteNT algorithm, and even the baseline can be regarded as a special case of AtteNT with a trivial sampling strategy. Therefore, Table 3 does not show “no effect of AtteNT”; instead, **it compares different choices of AtteNT hyperparameters**, and the **best configuration** (Incremental ratio, Incremental interval, Soft selection) reduces training time by 20.58% while improving all downstream tasks.
>
> Let us clarify the design of Table 3. The **Pre-training columns** report the upstream train-from-scratch stage, while the **Downstream columns** report performance on the downstream tasks. In the pre-training stage, all models are trained for the same 800 epochs, following the setup of Table 2. The **Ratio column** controls how the fraction of selected training samples changes over epochs. Following [3,4], the Cosine schedule increases the sampling ratio from 0.2 to 0.8 with a cosine schedule, while Incremental denotes a linear increase. The **Interval column** specifies how often we re-sample the subset: Fixed means we resample every epoch, whereas Incremental denotes an interval that grows linearly over training, here starting from 10 epochs. The **Selection column** describes how samples are chosen within each subset. Random denotes sampling without any difficulty prior, Hard denotes selecting only difficult samples according to AtteNT’s scoring, and **Soft denotes using the Gumbel-Top-k selection algorithm [5, 6], where the sampling probability is correlated with sample difficulty but even easy samples still have a small chance to be selected.
>
> The Downstream column then evaluates each pre-trained model on three tasks: **classification, semantic segmentation, and depth estimation** to jointly measure upstream efficiency and downstream quality. The purpose of this ablation is to understand how different components of AtteNT (ratio schedule, resampling interval, and selection strategy) affect both pre-training efficiency and downstream performance, and to identify a configuration that improves both.
>
> The key result appears in the fifth row of Table 3, where (Ratio, Interval, Selection) = (Incremental, Incremental, Soft). This configuration achieves **shorter training time while improving performance on all downstream tasks**. These results indicate that (i) starting with a small number of samples as a “warm-up,” (ii) gradually increasing both the sampling ratio and the resampling interval as training progresses, and (iii) using a soft difficulty-aware selection with Gumbel-Top-k can significantly boost upstream efficiency and downstream accuracy at the same time. We therefore adopt this configuration as our final AtteNT variant when comparing to the baseline, and show that our method does not require a trade-off between upstream efficiency and downstream performance, but instead achieves improvements in both.
>
> We acknowledge that these details are currently only described in Appendix C, which may have caused confusion. In the revision, we have updated the caption of Table 3 to make its intent clearer.
>
> - Table 3. Ablation study of AtteNT pre-training configurations. Pre-training reports the upstream train-from-scratch stage, while Downstream reports performance on classification, semantic segmentation, and depth estimation tasks. Ratio controls how the fraction of selected samples increases over epochs. Interval denotes how often the subset is re-sampled. Selection specifies the sampling strategy: Random (no difficulty prior), Hard (selects only difficult samples), and Soft (Gumbel-Top-k difficulty-aware sampling). The configuration (Incremental, Incremental, Soft) in the fifth row is adopted as our final AtteNT setting, as it simultaneously reduces pre-training time and improves performance on all downstream tasks.
>
> [4] Yoon J, Choi MK. "Exploring video frame redundancies for efficient data sampling and annotation in instance segmentation." CVPR 2023.
>
> [5] Kool W, Van Hoof H, Welling M. “Stochastic beams and where to find them: The Gumbel-top-k trick for sampling sequences without replacement.” ICML 2019.
>
> [6] Hao X, Hao J, Xiao C, Li K, Li D, Zheng Y. "Multiagent gumbel muzero: Efficient planning in combinatorial action spaces." AAAI 2024.
>
> > **[Q3]** Eq. 17 typo
>
> Thank you for the close reading. By the time we reach Equation 17 in Section 4.2, we have fully **shifted to the functional perspective**, in which the model evolves according to **functional gradient descent** with respect to the fixed canonical kernel $K(S_i, \cdot)$, rather than the time-varying ANTK kernel $K_{\theta_t}$. The use of $K$ reflects this analytical choice and is consistent throughout the functional analysis.

---

> > ### Comment · Reviewer_e8RB · 2025-11-25
> > **Response to Authors' Rebuttal**
> >
> > Thank you for the detailed response and clarification. Most of my questions are now resolved, and I believe that adding a point on potential future directions would be valuable for readers of the paper.
> > I do, however, have one remaining confusion after reading the authors’ rebuttal regarding Table 3. My understanding was that introducing the hard sampling based on AtteNT’s scoring was intended to be one of the main contributions of the paper. Yet, the results in Table 3 suggest that this sampling strategy does not consistently outperform the other approaches, despite the theoretical justification presented. I appreciate that, in practice, various factors may affect the setting and lead to deviations from the theoretical predictions, but I am curious whether the authors have any additional insight into why this might be the case.

---

> > > ### Author Response · Authors · 2025-11-26
> > >
> > > Thank you very much for the helpful suggestion on future work and the thoughtful follow-up. We are more than happy to resolve it.
> > >
> > > > On future work
> > >
> > > Following your suggestion, we have expanded the Concluding Remarks in the revision.
> > >
> > > > On the Hard vs. Soft selection in Table 3
> > >
> > > In Table 3, the deterministic Hard selection is theoretically the purest form of the AtteNT algorithm derived in our paper (the fastest 963 m). However, the Soft (Gumbel-Top-k) variant performs slightly better on most downstream tasks while remaining nearly as fast (980 m vs. 963 m).
> > >
> > > This is expected and desirable in large-scale training of attention models. Soft selection strictly follows the same AtteNT’s scoring as Hard, but allows the highest-scoring examples to be sampled with higher probability rather than deterministically. This small relaxation has two key benefits widely observed in large-scale attention training:
> > >
> > > 1. It preserves high diversity within each mini-batch (similar to how a modest temperature > 0 in next-token prediction prevents overly peaked distributions and improves training stability).
> > >
> > > 2. It naturally mitigates potential overfitting to outliers or noisy labels by preventing the model from repeatedly seeing exactly the same hardest subset across epochs, which keeps the effective curriculum smoother and more robust.
> > >
> > > As a result, Soft delivers the full acceleration guaranteed by our nonparametric teaching analysis while yielding better final performance across a broad range of vision and language tasks, which is exactly the same reason temperature is routinely used in language modeling.

---

> > > > ### Comment · Reviewer_e8RB · 2025-11-27
> > > > **Acknowledging Authors’ Rebuttal**
> > > >
> > > > Thank you for your response and changes to the paper.  I think the work is interesting and I will keep my positive score.

---

### Author Response · Authors · 2025-11-23
**Revision Note**

Dear Area Chair and Reviewers,

We have uploaded a revised version of the paper, in which all additional tables, figures, and experiments discussed in the rebuttal have been incorporated. In particular, we have added a new Appendix D, which includes additional experiments:
- D.1 Ablation of Sample Ratio
- D.2 Comparison to Established Methods
- D.3 Visualizing NTK Analysis

Best regards,

The Authors

---

### Author Response · Authors · 2025-11-29
**Author Final Remarks**

Dear Area Chair,

Thank you for handling our submission. We greatly appreciate the reviewers’ thorough and constructive feedback throughout the rebuttal phase. During the rebuttal, we had productive exchanges with the reviewers, and we would like to briefly summarize the outcome of these discussions.

## Summary of Rebuttal Discussions

- **Theoretical bridge and justification**: Reviewers appreciated the novelty of connecting attention learners with nonparametric teaching theory. After their questions, we provided additional intuition and an empirical NTK analysis (new Appendix D.3), showing that the kernel convergence underlying our theory indeed holds in realistic training regimes.

- **Practical concerns and experimental strengthening**: We clarified the measurement of training time, explained the robustness of AtteNT under label noise, and discussed how the selection ratio influences accuracy. The reviewers acknowledged that these points resolved their concerns. In response to their requests, we added ablations on sample ratio and included new comparisons with established sampling baselines. These experiments clarified the hyperparameters we selected and consistently support that AtteNT’s improvements stem from its principled design rather than generic heuristics.

- **Clarity and reproducibility**: Following reviewer suggestions, we substantially revised figure captions, expanded Experiments section, added a discussion of future work, and uploaded clearer code documentation. These changes greatly enhance readability and reproducibility.

## Summary of Contributions and Significance of AtteNT

We believe the revised manuscript now clearly conveys the following major advances:

- **Novel Paradigm**: AtteNT is the first framework that successfully extends nonparametric teaching theory to attention-based learners (transformers and ViTs), enabling theoretically grounded greedy example selection for sequence-to-property mappings.

- **Theoretical Contribution**: We analytically prove that parameter-based gradient descent in attention networks is asymptotically consistent with functional gradient descent in nonparametric teaching, showing that the dynamic Attention Neural Tangent Kernel (ANTK) converges pointwise to the importance-adaptive canonical kernel (Theorem 3). This closes a theoretical gap and, for the first time, justifies applying nonparametric teaching to modern attention architectures.

- **Practical Impact**: AtteNT reduces training time by an average of 13.01% when fine-tuning 7B-scale LLMs (LLaMA-2, Mistral, Gemma) and by 20.58% when pretraining ViTs from scratch (Multi-Modal MAE), while consistently preserving or improving downstream performance across language and vision tasks.

- **Broader Implications**: By bridging nonparametric teaching with the attention mechanism, AtteNT opens new research avenues at the intersection of machine teaching, kernel methods, and large-scale transformer training.

We believe these combined theoretical and empirical advances make AtteNT a meaningful contribution worthy of inclusion at ICLR.

Thank you again for the constructive and high-quality reviews.

Respectfully,

The Authors

---

### Meta-Review · Area_Chair_zJau · 2025-12-30

**Summary:**

The authors frame standard attention-based neural network training as implicitly performing a nonparametric version of machine teaching, where there are implicit weights given to training examples.  They show a connection between the parametric, standard learning framework, where parameters move to ensure loss is minimized, to the machine teaching function space view where examples are moved to ensure the prediction function is closer to the ground truth function.  They use this intuition to design a new learning algorithm, AtteNT, which upweights examples which have a large discrepancy between the predictions and true values.

The primary concerns of the reviewers concerned the restricted architecture (no skip conections, normalization, etc), asymptotic convergence instead of finite time, and experimental impact.  I find these to be reasonable assumptions for a theoretical paper.  The reviewers were largely in agreement that there was novelty developed in the understanding of learning dynamics of attention-based models, and how the nonparametric machine teaching lens was an interesting one and worth supporting.  I'm in agreement, and recommend acceptance.

**Reviewer Concerns:**

The reviewer concerns around experimental validation I think have largely been addressed.  I don't think it's necessary for improvement upon SoTA for this paper to be accepted.

The reviewer concerns regarding extension to multi-layer architectures and finite time bounds are still outstanding, but that is reasonable.  I think the question of noisy teacher labels would not necessarily be straightforward to address, but would be interesting, especially if the noise were adversarial.

**Reviewer Scores:**

I think it is likely that each reviewer maintained each of their scores, perhaps one would have increased it.  e8RB, jfuw, and gnaf each explicitly said that they would maintain their scores. The remaining reviewer csfv had concerns regarding experimental validation, code documentation, and clarity, which in my opinion were either addressed or relatively minor, in which case they may have raised to a 6 from 4.

---

### Decision · Program_Chairs · 2026-01-26

Accept (Poster)